# Neural Quantum States in Mixed Precision

**Massimo Solinas** [1]   **Agnes Valenti** [2]   **Nawaf Bou-Rabee** [3]   **Roeland Wiersema** [2]

## Abstract

Scientific computing has long relied on double precision (64-bit floating point) arithmetic to guarantee accuracy in simulations of real-world phenomena. However, the growing availability of hardware accelerators such as Graphics Processing Units (GPUs) has made low-precision formats attractive due to their superior performance, reduced memory footprint, and improved energy efficiency. In this work, we investigate the role of mixed-precision arithmetic in neural-network based Variational Monte Carlo (VMC), a widely used method for solving computationally otherwise intractable quantum many-body systems. We first derive general analytical bounds on the error introduced by reduced precision on Metropolis-Hastings MCMC, and then empirically validate these bounds on the use-case of VMC. We demonstrate that significant portions of the algorithm, in particular, sampling the quantum state, can be executed in half precision without loss of accuracy. More broadly, this work provides a theoretical framework to assess the applicability of mixed-precision arithmetic in machine-learning approaches that rely on MCMC sampling. In the context of VMC, we additionally demonstrate the practical effectiveness of mixed-precision strategies, enabling more scalable and energy-efficient simulations of quantum many-body systems.

[1]Fakultät für Informatik und Data Science, University of Regensburg, Universitätsstraße 31, D-93040, Regensburg [2]Center for Computational Quantum Physics, Flatiron Institute, 162 Fifth Avenue, New York, NY 10010, USA [3]Department of Mathematical Sciences, Rutgers University, Camden, New Jersey 08102, USA and Center for Computational Mathematics, Flatiron Institute, 162 Fifth Avenue, New York, NY 10010, USA. Correspondence to: Massimo Solinas <first1.last1@xxx.edu>, Roeland Wiersema <rwiersema@flatironinstitute.org>.

*Proceedings of the $43^{rd}$ International Conference on Machine Learning*, Seoul, South Korea. PMLR 306, 2026. Copyright 2026 by the author(s).

## 1. Introduction

Many problems in machine learning and the physical sciences require sampling from high-dimensional, often strongly multimodal probability distributions. Markov chain Monte Carlo (MCMC) methods are the standard approach, generating samples through local stochastic updates with guaranteed asymptotic convergence. In modern applications involving neural-network–parameterized distributions, the cost of generating sufficiently decorrelated MCMC samples often dominates the overall computational cost of inference or training (Neal, 1996; Papamarkou et al., 2022; Du & Mordatch, 2019; Du et al., 2021; Nijkamp et al., 2019; 2020; Samsonov et al., 2022).

In this work, we investigate a simple and broadly applicable strategy for reducing the computational cost of MCMC sampling: the use of reduced-precision arithmetic. Lower-precision formats, such as single precision (`f32`) and half precision (`f16` or `bf16`), have become standard in large-scale machine learning due to their substantial performance gains on modern hardware (Narayanan et al., 2021; Micikevicius et al., 2018; Sun et al., 2019). These gains, commonly benchmarked in terms of basic linear-algebra operations such as matrix multiplication (see Appendix Fig. 5), come at the expense of increased numerical error. Reliably assessing both speed-up and error that result from the use of reduced-precision formats is of notable interest in the field of scientific computing, where double precision has traditionally been employed to ensure accuracy (Higham & Mary, 2022; Haidar et al., 2018; Kashi et al., 2024). Here, we address this task for the use-case of MCMC sampling. Specifically, we derive theoretical bounds on the bias introduced by reduced precision in Metropolis–Hastings (MH) MCMC, and validate these bounds empirically. Finally, we demonstrate that substantial speed-ups can be achieved without loss of accuracy in a representative scientific application where MCMC constitutes a key computational bottleneck: neural quantum states (NQS) for the simulation of quantum many-body systems (Carleo & Troyer, 2017; Carrasquilla, 2020), a method which has been successfully applied to obtain state-of-the-art results on various open problems (Lange et al., 2024; Medvidović & Moreno, 2024).

The ability to efficiently simulate a quantum many-body system is crucial across a wide range of scientific disci-

plines. Its impact spans the discovery and characterization of quantum phases in physics, the development of deeper insights into molecular interactions and reaction mechanisms in quantum chemistry, and the design of novel materials with tailored properties in materials science. However, the exponential scaling of the degrees of freedom in quantum many-body wavefunctions poses a fundamental challenge for efficient simulation. NQS offer a prominent way to circumvent this challenge by compressing the quantum many-body wavefunction into a neural network and optimizing it toward the ground state of the system using variational Monte Carlo (VMC) (McMillan, 1965; Sorella, 2005). Rather than being data-driven, the approach proceeds by sampling the wavefunction at each optimization step via MCMC. Here we show that the MCMC sampling portion of the VMC algorithm can be significantly sped up with the use of lower-precision formats. We empirically demonstrate that this speed-up comes without sacrificing accuracy in the obtained ground-state approximation, and we relate these results to the derived theoretical bounds connecting them to the features of the targeted ground-state. Our findings show that reducing numerical precision can substantially accelerate NQS optimization while providing a rigorous framework to quantify low-precision sampling accuracy, with applications beyond NQS including Bayesian learning (Neal, 1996; Papamarkou et al., 2022) and energy-based models (Du & Mordatch, 2019; Du et al., 2021; Nijkamp et al., 2019; 2020; Samsonov et al., 2022).

**Our Contribution**

1. We derive bounds on the sampling error introduced by low-precision arithmetic on discrete MH MCMC. We construct a simplified toy model for sampling and find excellent agreement with the theoretical bounds.

2. We introduce a mixed-precision VMC framework that efficiently performs sampling in low precision while keeping the remaining computations in high precision.

3. We evaluate our method by training various models, including feed-forward networks and residual convolutional neural networks. Our results show that low-precision arithmetic can accelerate sampling by up to $3.5\times$ without degrading the training performance of the neural networks.

**Previous Work**  Perturbed Markov chains have been studied from several perspectives. Early work focused on robustness of ergodicity and convergence under small perturbations of the transition kernel, highlighting that seemingly benign numerical effects can destroy ergodicity in pathological cases (Roberts et al., 1998; Breyer et al., 2001). More recently, the effect of finite-precision arithmetic on the MH accept-reject step has been examined directly, with

an emphasis on how roundoff error can distort acceptance probabilities and degrade acceptance rates (Hoffman, 2020; Sountsov et al., 2024). A complementary line of work treats approximate Markov chains abstractly, providing general bounds on the difference between the invariant distribution of an exact chain and that of an approximate chain in terms of kernel-level perturbations and mixing properties (Johndrow et al., 2015; Johndrow & Mattingly, 2017). The effect of noise in a Markov chain need not always be destructive. In Pseudo-Marginal MCMC (Andrieu & Roberts, 2009; Nicholls et al., 2012; Liang & Jin, 2013; Medina-Aguayo et al., 2015), noise is intentionally introduced to accelerate Markov chain mixing.

The present work differs in scope and emphasis. Like other works, we consider a target distribution $\pi$ (corresponding to exact arithmetic) and a perturbed distribution $\tilde{\pi}$ arising from finite-precision effects in the evaluation of the log-density entering the MH accept–reject step. Rather than focusing on robustness of ergodicity or bounding error via abstract kernel norms, we analyze how finite-precision perturbations propagate through the MH acceptance rule and induce an asymptotic bias, in the sense of (Durmus & Eberle, 2024). Finally, the analysis of mixed-precision effects in the context of MCMC sampling employed in machine learning, or, NQS in particular, remains underdeveloped. The authors of (Chen & Heyl, 2024) mention using lower precision for the evaluation of the wave function, but the impact of this choice is still poorly understood.

## 2. Variational Monte Carlo

Physical properties at low temperature are dominated by the *ground state* of the physical system of interest. Determining the ground state amounts to finding the complex-valued eigenvector associated with the smallest eigenvalue (the *ground-state energy*) of a sparse Hermitian operator $H$, called the Hamiltonian, which encodes the total energy of the system. We denote this eigenvector by $|\psi\rangle$ and refer to it as the *ground-state wavefunction*. Throughout this manuscript, we consider an effective model consisting of spin-$1/2$ quantum degrees of freedom on a lattice of $N$ sites. The associated Hilbert space has dimension $2^N$; thus, $|\psi\rangle \in \mathbb{C}^{2^N}$ and the Hamiltonian $H$ is a $2^N \times 2^N$ matrix, making direct diagonalization computationally prohibitive.

Within VMC, this exponential scaling is avoided by approximating the ground state using two crucial ingredients: (i) a parameterized representation of the quantum many-body wave-function $|\psi_\theta\rangle \in \mathbb{C}^{2^N}$; and, (ii) optimization of the parameters $\theta$ by minimizing the Rayleigh-Ritz quotient $E_\theta = \langle\psi_\theta| H |\psi_\theta\rangle / \langle\psi_\theta|\psi_\theta\rangle$, which provides an upper bound on the ground-state energy. For (i), we make use of NQS. In particular, an (unnormalized) quantum many-body

wavefunction $|\psi_\theta\rangle$ can be decomposed as

$$|\psi_\theta\rangle = \sum_{x \in \mathcal{X}} \psi_\theta(x) |x\rangle, \quad \mathcal{X} = \{0,1\}^N, \qquad (1)$$

where $\psi_\theta : \mathcal{X} \to \mathbb{C}$, $\theta \in \mathbb{R}^{N_\theta}$, assigns a complex amplitude to each standard basis vector, indexed by the binary vector $x$, corresponding to a spin-configuration. A generic parameterization of this quantum many-body wave-function can be obtained by representing the function $\psi_\theta$ as a neural network. Common choices of architecture include Restricted Boltzmann Machines (RBMs) (Carleo & Troyer, 2017), Convolutional Neural Networks (CNNs) (Choo et al., 2019), Recurrent Neural Networks (RNNs) (Hibat-Allah et al., 2020) and Transformers (Viteritti et al., 2023).

For (ii), we need to efficiently estimate the loss function:

$$E_\theta = \frac{\sum_{x,x' \in \mathcal{X}} \psi_\theta^*(x)\psi_\theta(x')H(x,x')}{\sum_{x \in \mathcal{X}} |\psi_\theta(x)|^2} = \mathbb{E}_{x \sim \pi_\theta}\left[E_\theta^l(x)\right] \tag{2}$$

where $H(x,x')$ are the (sparse) matrix elements of the Hamiltonian and $\pi_\theta(x) = p_\theta(x)/\sum_{x \in \mathcal{X}} p_\theta(x)$, with $p_\theta(x) = |\psi_\theta(x)|^2$. Moreover we defined the "local energies" as:

$$E_\theta^l(x) = \sum_{x' \in C(x)} H(x,x')e^{\log\psi_\theta(x') - \log\psi_\theta(x)}, \tag{3}$$

where the set $C(x)$ contains all configurations that have a non-zero contributions when $H$ is applied to $x$. In typical physical systems $|C(x)|$ scales at most polynomially in $n$, which makes evaluating the local energies efficient.

In order to evaluate Eq. 2 exactly, we require computing the wavefunction on a number of configurations that is exponential in the system size $N$. For this reason, the expectation $\mathbb{E}_{x \sim \pi_\theta}\left[E_\theta^l(x)\right]$ is typically approximated using MCMC sampling. The same sampling procedure is used to estimate the gradient of the loss function (see Section I.1). To this end, MH is employed to approximately sample from $\pi_\theta$. In practice, the neural network directly parametrizes $\log\psi_\theta(x)$, from which the unnormalized distribution $p_\theta$ is obtained via $\log p_\theta(x) = 2\Re[\log\psi_\theta(x)]$.

From a machine learning perspective, VMC can be interpreted as an unsupervised learning approach, where the training dataset, consisting of feature vectors (spin-configurations) $x$, is generated dynamically at each optimization step by sampling from the current neural network itself via MCMC. Accordingly, a single VMC optimization step comprises three main components: (i) sampling configurations; (ii) computing gradients via backpropagation; and, (iii) applying gradient preconditioning, which is essential for stable and accurate convergence to the ground state. In

the following, we examine the effect of mixed precision on (i), the sampling component of VMC.

## 3. Mixed Precision During Sampling

Because each VMC iteration requires repeated evaluation of the wave-function amplitudes, VMC remains computationally intensive even on modern GPUs (Rende et al., 2024; Roth et al., 2025). We therefore study the effect of reduced-precision arithmetic on the sampling component of VMC. All proofs are deferred to Section C.

Let $p_\theta$ denote the exact target distribution, and write $\pi(x) = p_\theta(x)$, suppressing the dependence on parameters $\theta$ for notational simplicity. Reduced precision is modeled abstractly as an additive perturbation of the log-probability,

$$\log\pi(x) \to \log\pi(x) + \delta(x), \tag{4}$$

where $\delta(x)$ represents the cumulative numerical error arising from finite-precision evaluation of $\psi_\theta(x)$ and the subsequent computation of $\log\pi(x)$. Since the perturbation does not preserve the normalization, we define the resulting normalized probability distribution $\tilde\pi(x)$ as

$$\tilde\pi(x) = \frac{1}{\mathcal{Z}}\pi(x)e^{\delta(x)}, \quad \mathcal{Z} = \sum_{x \in \mathcal{X}} \pi(x)e^{\delta(x)}. \tag{5}$$

To quantify the difference between the true and perturbed distributions, we utilize the *total variation* (TV) distance,

$$\|\pi - \tilde\pi\|_{\text{TV}} := \frac{1}{2}\sum_{x \in \mathcal{X}} |\pi(x) - \tilde\pi(x)|, \tag{6}$$

a standard metric for comparing probability measures (Meyn & Tweedie, 1993; Levin & Peres, 2017; Douc et al., 2018).

When the target distribution is perturbed, the expectation of any observable $f : \mathcal{X} \to \mathbb{R}$ estimated via Monte Carlo, such as the VMC energy in Eq. 2, generally differs from its expectation under the true target. For bounded functions $f$ satisfying $\max_x |f(x)| \le A$, this bias satisfies

$$|\mathbb{E}_{x \sim \tilde\pi}[f(x)] - \mathbb{E}_{x \sim \pi}[f(x)]| \le 2A\|\pi - \tilde\pi\|_{\text{TV}}, \tag{7}$$

which follows directly from the triangle inequality and the definition of total variation distance. In VMC, energy gradients are estimated from samples drawn from $\pi(x)$; therefore, understanding how deviations from the true distribution affect expectation values is crucial, as gradient noise can impact the optimization. In order to practically estimate these effects, we derive concrete bounds on the RHS of Eq. 7, that directly relate to the moments of the perturbation $\delta$.

To obtain a baseline bound on the bias induced by finite-precision arithmetic, we model the perturbation $\delta$ as Gaussian. This assumption is motivated by the central limit theorem and supported empirically in Section 4. For a coarse

bound, we deliberately ignore the dynamics of the sampler and focus only on how the perturbation alters the target distribution. In particular, if $\delta \sim \mathcal{N}(\mu, \sigma^2)$, we obtain

$$\mathrm{KL}(\pi \| \tilde{\pi}) = \log \mathbb{E}_{x \sim \pi}[e^{\delta(x)}] - \mathbb{E}_{x \sim \pi}[\delta(x)] = \frac{\sigma^2}{2}. \quad (8)$$

By Pinsker's inequality, this implies

$$\|\pi - \tilde{\pi}\|_{\mathrm{TV}} \leq \frac{1}{2}\sigma. \quad (9)$$

Incorporating the mixing behavior of the underlying Markov chain yields substantially sharper bounds in regimes where the chain mixes rapidly.

### 3.1. Properties of the Finite-Precision Markov Chain

By taking into account the mixing time of the Markov chains, we show that it is possible to derive a tighter bound than the one in Eq. 9. To this end, we introduce a quantitative notion of convergence to stationarity and relate it to spectral properties of the transition kernel. Background on spectral gaps, contraction in total variation, and their relationship to mixing times is provided in Section B.

For a target distribution $\pi(x)$ on a finite state space $\mathcal{X}$, the MH acceptance probability $\alpha(x, y)$ for a proposed update $x \to y$, under a proposal distribution $q(y \mid x)$, is

$$\alpha(x, y) = \min\{1, s(x, y)\}, \quad (10)$$
$$s(x, y) := \exp\big(\log \pi(y) - \log \pi(x)\big) \quad \forall x, y \in \mathcal{X}.$$

The resulting Markov chain is fully specified by the stochastic transition matrix

$$P(x, y) = \begin{cases} q(y \mid x)\alpha(x, y) & \text{if } x \neq y, \\ 1 - \sum_{z \neq x} P(x, z) & \text{if } x = y. \end{cases}$$

A central result in Markov chain theory states that, under mild regularity conditions (irreducibility and aperiodicity), the distribution of the chain converges exponentially fast to its stationary distribution $\pi$, uniformly over all initial states (Meyn & Tweedie, 1993; Douc et al., 2018).

While convergence rates are formally governed by the spectral gap of $P$, computing this gap is computationally intractable for the large state spaces encountered in NQS. Instead, we utilize the Doeblin minorization condition (Douc et al., 2018) to provide a robust analytical framework. Specifically, if there exists a constant $\xi \in (0, 1)$ and a probability distribution $\nu$ such that $P(x, \cdot) \geq \xi\nu(\cdot)$ for all $x \in \mathcal{X}$, then $P$ acts as a strict contraction in TV distance, so that for all probability distributions $\mu$ and $\mu'$:

$$\|\mu P - \mu' P\|_{\mathrm{TV}} \leq r\|\mu - \mu'\|_{\mathrm{TV}}, \quad (11)$$

where $r := 1 - \xi$ is the contraction rate. The quantity $(1 - r)^{-1}$ serves as a proxy for the mixing time, characterizing how rapidly the chain converges to its stationary distribution $\pi$. On a finite state space, this contractivity ensures that all non-trivial eigenvalues of $P$ have a modulus no greater than $r$, thereby guaranteeing a strictly positive spectral gap. The utility of the Doeblin condition in our context is twofold: it provides a lower bound on the spectral gap without requiring direct spectral analysis, and more importantly, it establishes the stability of the chain under perturbation. By assuming our chains satisfy this condition, we can rigorously quantify how the long-run sampling distribution is biased when the transition kernel is perturbed by the noise inherent in reduced-precision arithmetic.

The perturbed MH acceptance probability associated to $\tilde{\pi}(x)$, as defined in Eq. 4, can be written as:

$$\tilde{\alpha}(x, y) = \min\left\{1, s(x, y)e^{\varepsilon(x,y)}\right\}, \quad (12)$$

where $\varepsilon(x, y) := \delta(y) - \delta(x)$. This, combined with the proposal distribution $q(y \mid x)$ defines the perturbed Markov kernel $\tilde{P}$. Although it is known that numerical perturbations can, in pathological cases, destroy ergodicity (Roberts et al., 1998; Breyer et al., 2001), we restrict attention to settings in which the perturbed kernel $\tilde{P}$ remains irreducible and aperiodic, and therefore admits a unique stationary distribution $\tilde{\pi}$. In this regime, when the ideal Markov chain mixes quickly ($r \ll 1$), small perturbations of its transition kernel cannot push the stationary distribution very far. We formalize this next.

**Theorem 3.1.** *Let $\pi$ and $\tilde{\pi}$ be probability measures on $\mathcal{X}$. Let $P$ and $\tilde{P}$ be Markov kernels with invariant distributions $\pi$ and $\tilde{\pi}$, respectively. Assume that $P$ is a strict contraction in TV with constant $r \in (0, 1)$ in the sense of Eq. 11. Then*

$$\|\tilde{\pi} - \pi\|_{\mathrm{TV}} \leq \left\|\tilde{\pi}(\tilde{P} - P)\right\|_{\mathrm{TV}} \frac{1}{(1-r)}.$$

The proof follows from the triangle inequality together with the invariance relations $\pi P = \pi$ and $\tilde{\pi}\tilde{P} = \tilde{\pi}$, i.e., $\pi$ and $\tilde{\pi}$ are left eigenvectors with eigenvalue 1 of $P$ and $\tilde{P}$, respectively. Theorem 3.1 reduces control of the bias in the stationary distribution to control of the difference between the Markov kernels:

$$\Delta P = \begin{cases} q(y \mid x)\Delta\alpha(x, y) & \text{if } x \neq y, \\ -\sum_{z \neq x} q(z \mid x)\Delta\alpha(x, z) & \text{if } x = y, \end{cases} \quad (13)$$

where we defined $\Delta P := \tilde{P} - P$ and $\Delta\alpha := \tilde{\alpha} - \alpha$. The sensitivity of the Markov kernel to finite-precision errors can be qualitatively understood by analyzing $|\Delta\alpha(x, y)|$ as a function of the acceptance ratio $s(x, y)$ and the error difference $\varepsilon(x, y)$. In the following discussion we assume


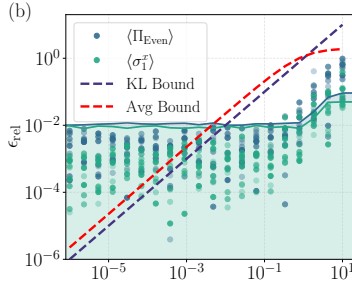
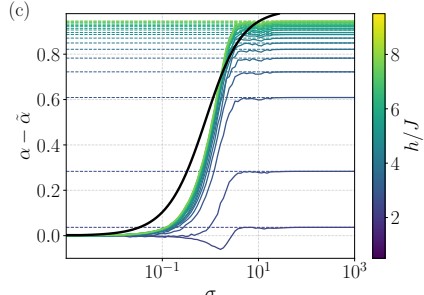

*Figure 1.* Panel (a) displays the acceptance difference $|\Delta\alpha(x,y)|$ as a function of the acceptance ratio $s(x,y)$ and the error difference $\varepsilon(x,y)$. The solid red line corresponds to $s(x,y) = e^{-\varepsilon(x,y)}$. For $\varepsilon(x,y) < 0$ the plot is inverted over both the x-axis and y-axis. Panels (b) and (c) present results for the noisy RBM. Panel (b) shows the relative error between expectation values computed with the unperturbed and noisy RBM as a function of noise parameter $\sigma$ compared to the KL bound of Eq. 9 and the average bound of Eq. 14. The expected values of both the Pauli-X operator $\sigma^x$ and the projector onto the even bitstrings $\Pi_{\text{Even}}$ are investigated. Individual points correspond to different random initializations, while shaded regions indicate confidence intervals determined by Monte Carlo sampling noise, calculated as $3\sqrt{\text{Var}[\epsilon_{\text{rel}}]/N_{\text{samples}}}$, where the factor of 3 corresponds to approximately a 99.7% confidence interval assuming a Gaussian distribution of the noise. Results are obtained using $2^{18}$ samples, with the average bound computed for $r = 0$. Panel (c) shows the acceptance rate as a function of $\sigma$ for different values of $h/J$ in the TFIM. The dark line represents the bound for $|\Delta\alpha|$, while the dashed lines correspond to the value of the unperturbed acceptance rate $\alpha$.

$\varepsilon(x,y)$ to be positive, as the negative case is symmetric. As illustrated in Fig. 1(a), three distinct regimes emerge. When $s(x,y) \le e^{-\varepsilon(x,y)}$, both acceptance probabilities scale proportionally to $s(x,y)$, so the effect of finite-precision error is suppressed in low-acceptance regimes. Conversely, for $s(x,y) \ge 1$, both chains accept with probability one, rendering the error irrelevant. The kernel is most sensitive in the intermediate regime, $e^{-\varepsilon(x,y)} < s(x,y) < 1$, where proposals lie near the acceptance threshold ($s \approx 1$). Importantly, since $\Delta P$ is weighted by the proposal distribution $q$, the distance between the two kernels is significant only if such sensitive transitions are proposed with non-negligible probability $q(y \mid x)$. This observation, leads us to the following:

**Theorem 3.2.** *Let $\mathcal{X}$ be a finite state space, and let $P$ and $\tilde{P}$ be MH kernels on $\mathcal{X}$ constructed with the same proposal distribution $q(y \mid x)$ and acceptance probabilities $\alpha$ and $\tilde{\alpha}$, as defined in Eqs. 10 and 12, respectively. Then, for any probability distribution $\pi'$ on $\mathcal{X}$,*

$$\left\|\pi'(\tilde{P} - P)\right\|_{\text{TV}} \le 1 - \sum_{y \in \mathcal{X}} \mathbb{E}_{x \sim \pi'}\left[q(y \mid x)e^{-|\varepsilon(x,y)|}\right].$$

The proof bounds $\|\pi'(\tilde{P} - P)\|_{\text{TV}}$ by first controlling the row-wise $\ell^1$ difference between $\tilde{P}(x,\cdot)$ and $P(x,\cdot)$ in terms of the acceptance–probability difference $|\Delta\alpha(x,y)|$, taking care of the diagonal term via conservation of probability. A case analysis of the MH acceptance function then yields $|\Delta\alpha(x,y)| \le 1 - e^{-|\varepsilon(x,y)|}$ for $x \ne y$, and averaging against the proposal $q(y \mid x)$ gives the stated bound.

The above bounds do not require any assumptions on the distribution of error term $\delta(x)$, the proposal distribution $q(y \mid x)$, or the target distribution $\pi(x)$; consequently, Theorem 3.1 and Theorem 3.2 are completely general. In the

following we derive a bound tailored to VMC, by focusing on local MH proposals and, as in the baseline bound in Eq. 9, assuming that the error is Gaussian distributed.

**Theorem 3.3** (Local MH with Gaussian Noise). *Let $\pi$ and $\tilde{\pi}$ be probability measures on the discrete space $\mathcal{X} = \{0,1\}^N$, related by the perturbation $\delta$ defined in Eq. 5. Let $P$ and $\tilde{P}$ be the MH kernels targeting $\pi$ and $\tilde{\pi}$, respectively, constructed using a symmetric single-bit-flip proposal distribution $q(y \mid x)$. Assume that $P$ is a strict contraction in TV with constant $r \in (0,1)$, satisfying Eq. 11. Furthermore, assume that for a joint draw $X \sim \tilde{\pi}$ and $Y \sim q(\cdot \mid X)$, the log-density increment $\varepsilon(X,Y) := (\delta(Y) - \delta(X)) \sim \mathcal{N}(\mu, 2\sigma^2)$. Then,*

$$\|\tilde{\pi} - \pi\|_{\text{TV}} \le \frac{1}{1-r}\left(1 - \frac{e^{\sigma^2}}{2}\left[e^{-\mu}\operatorname{erfc}\left(\sigma - \frac{\mu}{2\sigma}\right)\right.\right.$$
$$\left.\left. + e^{\mu}\operatorname{erfc}\left(\sigma + \frac{\mu}{2\sigma}\right)\right]\right). \tag{14}$$

The bound in Theorem 3.3 highlights a clear separation of effects. The bias induced by mixed-precision arithmetic is controlled by two quantities: (i) the noise level $\sigma$ arising from finite-precision evaluation of the log density, and (ii) the contraction rate $r$ of the ideal Markov chain. When the ideal chain mixes rapidly ($r \ll 1$), small numerical perturbations cannot significantly distort the invariant distribution. Conversely, slow mixing amplifies the effect of numerical noise. Importantly, the bound depends on $\sigma$ but not explicitly on the system size $N$, except through its influence on $\sigma$ and $r$. This clarifies that mixed-precision sampling is viable precisely in regimes where the sampler is already well behaved.

The proof follows by evaluating the Gaussian integral in

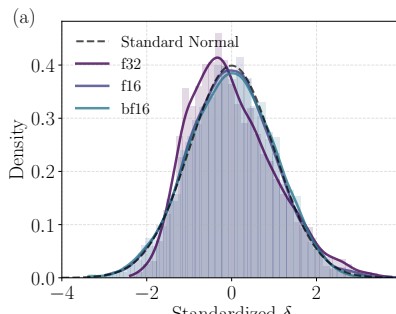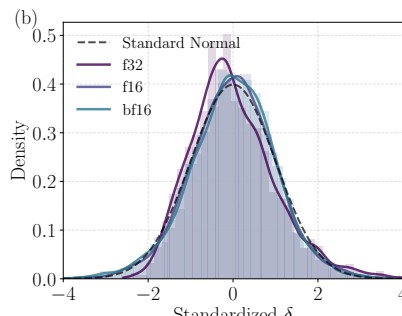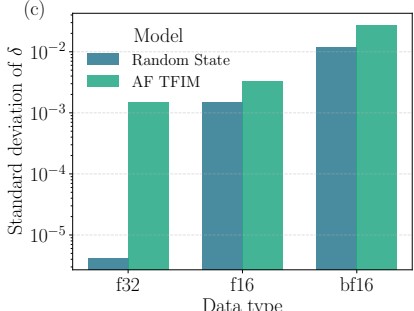

*Figure 2.* Standardized law of $\delta$ for a random state (panel (a)) and for the ground state of the TFIM in the antiferromagnetic phase at $h/J = 0.5$ (panel (b)). The distributions are computed for different data types on a spin chain of length $N = 12$, using all configurations of the Hilbert space as defined in Eq. 4. Panel (c) shows the standard deviation of each $\delta$ distribution of the states in panels (a) and (b).

Theorem 3.2 and combining the result with Theorem 3.1. In the case of centered noise, i.e. $\mu = 0$, we obtain

$$\|\tilde{\pi} - \pi\|_{\mathrm{TV}} \leq \frac{1 - e^{\sigma^2} \operatorname{erfc}(\sigma)}{(1 - r)}. \tag{15}$$

As $\sigma \to 0$, the right hand side scales as $\sigma/(1-r)$. While the bound in Eq. 15 scales as the one provided in Eq. 9, it provides a more accurate description of the sampling process by incorporating the contraction rate $r$, and therefore the autocorrelation and thermalization time of the sampler, into the analysis. For VMC, this means that as long as the MH sampler mixes sufficiently rapidly and the mixed-precision noise $\sigma$ is small, the resulting bias scales as $O(\sigma^2)$, instead of $O(\sigma)$. Our analysis is carried out in high-dimensional state spaces but assumes local MH proposals, so that the perturbation enters only through the log-density increment $\varepsilon(x, y)$ between neighboring states. This locality suggests that any growth of the error with system size is likely to be mild.

## 4. Numerical Results

To assess the practical relevance of the derived bounds, we conduct a numerical study of how low-precision arithmetic influences VMC. In the following, we treat double precision (f64) as numerically exact. Moreover, in all optimizations, we employ stochastic reconfiguration to precondition the gradients, which accelerates convergence and yields a more accurate ground-state approximation (see Section I.2 for further details).

### 4.1. Noisy Restricted Boltzmann Machine

As a warmup example, we consider a commonly used neural network ansatz in VMC: the RBM (see Appendix G). To gain precise control over the perturbation scale $\sigma$, we inject

Gaussian noise at the output of the model:

$$\log \tilde{\psi}_\theta(x) = \log \psi_\theta(x) + \zeta(x)/2 \tag{16}$$

with $\zeta(x) \sim \mathcal{N}(0, \sigma)$. The factor $1/2$ ensures that resulting log-density difference $\pi_\theta(x)$ has variance $\sigma$. Crucially, to mimic the type of error introduced by reduced precision, the RBM is designed such that for a given input $x$, the realization $\zeta(x)$ is held fixed across evaluations. In panel (b) of Fig. 1, we study the different bounds in practice. Importantly, we observe that, for small $\sigma$, the Monte Carlo sampling error in the relative difference between expectation values computed from the perturbed and unperturbed distributions exceeds the derived bounds. Consequently, until $\sigma$ becomes of order unity, the sampling procedure is expected to be effectively insensitive to the perturbation. As a second step, we train several NQS using an RBM without noise. Specifically, we consider the transverse-field Ising model (TFIM), described by the Hamiltonian

$$\hat{H} = J \sum_{\langle i,j \rangle} \hat{\sigma}_i^z \hat{\sigma}_j^z + h \sum_i \hat{\sigma}_i^x, \tag{17}$$

where $\hat{\sigma}^x$ and $\hat{\sigma}^z$ are Pauli matrices and $J, h \in \mathbb{R}$. Here we focus on the antiferromagnetic regime, $J > 0$. This model captures the competition between an ordering tendency that favors an alternating up–down pattern, driven by the $J$ term, and quantum fluctuations induced by the transverse field $h$, which mix spin configurations and promote superposition states. We use this property of the system to generate a family of NQSs with different structural properties, by obtaining its ground state for various values of the transverse field $h/J$. In particular, for $h/J \to 0$, the squared modulus of the wavefunction is sharply concentrated around the two Néel configurations (antiferromagnetic phase), whereas increasing $h/J$ spreads the probability distribution over many spin configurations, approaching a nearly flat probability distribution for $h/J \to +\infty$ (paramagnetic phase).

Once all NQSs are trained and have converged, we use the

learned parameters to initialize the noisy RBMs and perform sampling with different values of $\sigma$. As illustrated in panel (c) of Fig. 1, this procedure allows us to systematically study how the MH acceptance rate depends on the variance of the noise for qualitatively different quantum states that are usually encountered in practice. At the baseline (dashed lines), different wavefunctions exhibit distinct acceptance rates; sharper probability distributions generally yield lower rates, as the MH algorithm finds fewer states with significant transition probabilities. While $|\Delta\alpha|$ vanishes for small $\sigma$, it approaches $\alpha$ as the noise increases, indicating complete suppression of the acceptance rate. Moreover, once this regime is reached, the derived bound is violated, indicating that the ergodicity assumption no longer holds. We can therefore identify two distinct regimes. In the small-$\sigma$ regime, wavefunctions trained on systems with larger values of $h/J$, thus with a broader shape, are affected by noise at smaller $\sigma$, confirming Theorem 3.2. On the other hand, in the large-$\sigma$ regime, the perturbed distribution becomes concentrated around a small number of randomly selected configurations, effectively obscuring the underlying true distribution. Here, the perturbation is sufficiently strong to inhibit thermalization of the Markov chains, resulting in a loss of ergodicity. Notably, for $h/J < 1$, $\tilde{\alpha}$ exhibits a slight increase before eventually dropping to zero. This bump arises as, for a narrow window of noise strengths, the noise occasionally generates a small number of additional configurations whose probabilities become comparable to that of the true underlying one, temporarily boosting the acceptance rate. However, once $\sigma$ becomes large enough, the acceptance rate falls again, as these noise-induced configurations overwhelmingly dominate over the true state. This effect appears only in the regime $h/J \simeq 0$, where the underlying true distribution is already sharply peaked, making the brief increase in competing configurations sufficiently pronounced to produce a visible change in the acceptance rate.

### 4.2. Practical example in low precision

Before performing full variational VMC calculations with reduced-precision sampling, we examine the distribution of $\delta$ for two representative wavefunctions. This allows us to empirically verify the validity of the Gaussian approximation and determine whether $\sigma$ remains sufficiently small within the context of our model. We consider a randomly initialized state, corresponding to a flat distribution, and the ground state of the TFIM in the antiferromagnetic phase. As illustrated in Fig. 2, the errors at the output of the different models approximately follow a Gaussian distribution. To corroborate this observation, we perform the Shapiro–Wilk test (Shapiro & Wilk, 1965), obtaining statistics $\geq 0.97$ for all distributions, which confirms near-normality. More importantly, the variance of $\delta$ is consistently smaller than 1,

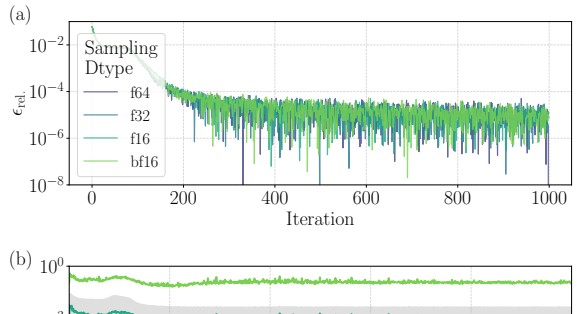

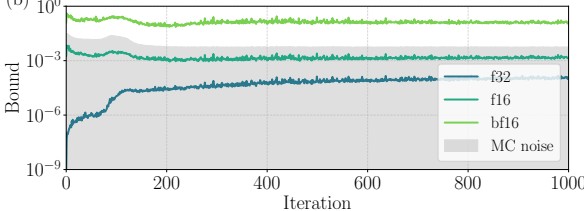

*Figure 3.* Panel (a) shows the relative energy error with respect to the double-precision reference solution as a function of training step. Optimization is performed using the mixed-precision scheme introduced in this work, with sampling carried out in reduced precision. We use a ResCNN with kernel size $(3, 3)$, four residual blocks, and 16 filters to study the two-dimensional TFIM at the critical point with linear system size $L = 10$ $(N = L^2)$. Panel (b) reports, for the same optimization, the minimum between the KL bound in Eq. 9 and the bound in Eq. 14, evaluated on the absolute difference between gradients from perturbed and unperturbed distributions. Shaded regions indicate the Monte Carlo confidence interval, given by $3\sqrt{2}\sqrt{\mathrm{Var}[\nabla E_\theta^l(x)]/N_{\mathrm{samples}}}$, where $\sqrt{2}$ accounts for error propagation in the gradient difference and the factor of 3 corresponds to a 99.7% Gaussian confidence level. For this experiment, we measure total training speedups of $2.057\times$ for `f32`, $2.296\times$ for `f16`, and $2.257\times$ for `bf16`.

indicating that MCMC should be able to converge reliably to the underlying distribution despite the reduced precision. In principle, $\sigma$ itself could depend on the size of $\mathcal{X}$; however, we find no such dependence empirically (see Section D).

To test whether VMC can be carried out in mixed precision, we design a dedicated optimization scheme that enables sampling in reduced precision while ensuring stable training. The key idea is to maintain two separate copies of the model throughout the training process. The first copy operates entirely in double precision and is responsible for both the forward and backward passes, ensuring accurate gradient computation and parameter updates. At each iteration, once the parameters are updated in this high-precision model, they are downcast to the target lower precision and transferred to the second copy of the model. This low-precision model is then used exclusively for sampling new configurations according to the probability distribution defined by the squared modulus wavefunction.

To assess the applicability of our proposed optimization scheme, we turn to a more challenging problem. We take the two dimensional TFIM for a square lattice with linear

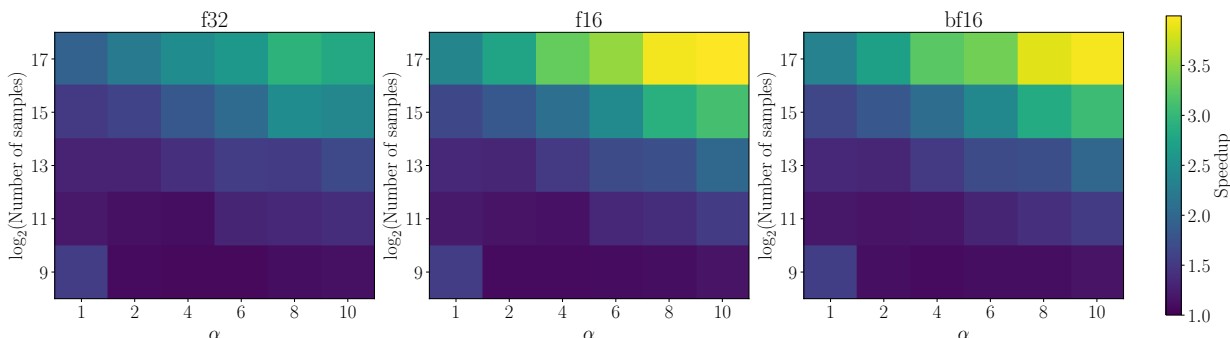

*Figure 4.* Sampling speedup obtained on a NVIDIA H100 GPU with RBMs with different data types. Results are shown as a function of both the number of samples $N_s$, with the number of Markov chains set to $N_s/4$, and the density of parameters $\alpha$, defined as the number of parameters per number of spins. The sampling speed-up is defined as the ratio of the double-precision sampling execution time to the reduced-precision execution time.

dimension $L = 10$ so $N = L^2$, and approximate the ground state at the critical point (Blöte & Deng, 2002), where the state is much more challenging to learn. For this task we adopt a ResCNN (Barton et al., 2026) architecture with 4 residual blocks, 16 filters and a $(3 \times 3)$ kernel (see Appendix H). Throughout the optimization, we compare the resulting energy to that obtained from a previously trained double-precision optimization. As shown in Fig. 3 (a), the relative error as a function of training steps is nearly identical across all data types, providing further evidence that sampling in lower precision does not compromise the final accuracy of the VMC calculation. To connect the theoretical error bounds with their practical impact during optimization, we perform an additional experiment. Since gradients are estimated via MC sampling, they are affected by both stochastic sampling noise and errors due to reduced numerical precision. A sufficient, though not necessary, condition for stable optimization is that the MC noise dominates, or is comparable to, the low-precision error. Accordingly, at each optimization step we measure the MC gradient noise and the standard deviation $\sigma$ of the low-precision error, which enters the derived upper bounds on $|\mathbb{E}_\pi[\nabla E_\theta^l] - \mathbb{E}_{\tilde{\pi}}[\nabla E_\theta^l]|$, where $E_\theta^l(x)$ is defined in Eq. 2. As shown in Fig. 3 (b), the MC noise exceeds the bound throughout training when using single precision and f16, explaining the unchanged optimization dynamics. This clearly explains why the optimization dynamics are not affected in these cases. By contrast, this separation is not observed for bf16. Nevertheless, as discussed previously, this does not imply that the optimization must fail: the gradients can tolerate a wider range of noise levels before the optimization process is effectively compromised. However, since the speedups observed for f16 and bf16 are comparable, one can safely use f16 for sampling without affecting the optimization. In Section F, we repeat the experiment for the antiferromagnetic and paramagnetic phases of the TFIM, and in Section E for the Heisenberg model. In all cases, we find that $\sigma$ remains sufficiently small so as not to compromise training.

Finally, we analyze the speedup achieved by sampling with an RBM as a function of both the number of parameters and the number of samples, while fixing the number of samples per chain to 4. The results presented in Fig. 4 demonstrate a clear trend: the achieved speedup systematically increases with both the number of samples and the number of model parameters. The mechanisms underlying these two dependencies, however, are distinct. In the first case, increasing the number of samples effectively enlarges the batch of configurations that the model processes simultaneously at each iteration. This higher workload per call allows the GPU to operate closer to its memory capacity, thereby improving hardware utilization and maximizing the achievable throughput. In contrast, increasing the number of parameters affects the computational load rather than the memory saturation. A larger parameter space increases the number of floating-point operations required whenever the model is evaluated. Since modern GPUs are designed to perform massive numbers of such operations in parallel, scaling up the parameter count of the neural network drives the computation further into the regime where the GPU's arithmetic units can be fully leveraged. In Section H we present the same experiment carried out with the ResCNN, where we also find large speedups for the sampling.

## 5. Conclusion

In this work, we derived theoretical bounds on the bias introduced by reduced-precision MH MCMC and demonstrated their relevance in a scientific application to NQS. For typical VMC optimizations, we find that the resulting sampling bias is small enough to leave convergence to the ground-state minimum essentially unchanged, while delivering substantial speedups. Our analysis in Theorem 3.3 relies on a particular noise model and proposal distribution. Extending the same approach to richer proposal families and more complex MCMC dynamics should be feasible, and we expect similar tools to carry over with only modest

additional effort. Although mixed-precision sampling can provide meaningful gains in large-scale simulations, even larger improvements may come from accelerating the computation of local energies (cf. Eq. 2) and the VMC preconditioning step. We briefly explore this direction in Section I and highlight several practical challenges that arise. Mixed-precision linear algebra has advanced rapidly in recent years (Higham & Mary, 2022), including mixed-precision iterative solvers (Haidar et al., 2018), and these developments are promising for VMC workflows. We tested whether such solvers could substantially accelerate the preconditioning step, and observed notable benefits only for very large systems (matrices larger than $2^{15} \times 2^{15}$). To support further exploration, we provide a JAX interface at (Wiersema, 2025). In conclusion, while mixed precision is now standard practice in machine learning, our results suggest it remains an underexplored, and potentially high-impact, tool in MCMC-driven scientific computing, especially as more workflows increasingly depend on GPU hardware.

All code used to generate the data presented in this work is publicly available at `https://github.com/mso linas28/Neural_Quantum_States_in_Mixed _Precision`. Our implementations are based on NetKet (Vicentini et al., 2022) and JAX (Bradbury et al., 2018).

## Impact Statement

We introduce a methodology for performing optimization with reduced numerical precision in sampling-based computational techniques, with a specific focus on NQS together with VMC. This approach addresses the escalating environmental impact of large-scale simulations by cutting energy and resource requirements. By making high-fidelity quantum simulations more computationally affordable, we broaden participation in the field and accelerate the application of machine learning to critical challenges in materials science and quantum chemistry.

## Acknowledgements

RW and AV acknowledge support from the Flatiron Institute. The Flatiron Institute is a division of the Simons Foundation.

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

## A. Data types

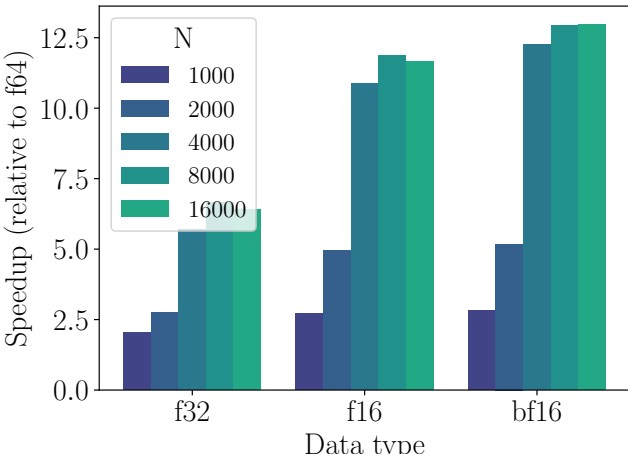

*Figure 5.* Speedup over different data types relative to double precision, obtained by multiplying two matrices of shape $(N, N)$ on an NVIDIA H100 GPU. Each result is obtained by averaging over 50 matrix multiplications.

| Name | Exp. | Signif. | Range | Error |
|------|------|---------|-------|-------|
| `bf16` | 8 | 7 | $10^{\pm 38}$ | $3.91 \times 10^{-3}$ |
| `f16` | 5 | 10 | $10^{\pm 5}$ | $4.88 \times 10^{-4}$ |
| `f32` | 8 | 23 | $10^{\pm 38}$ | $5.96 \times 10^{-8}$ |
| `f64` | 11 | 52 | $10^{\pm 308}$ | $1.11 \times 10^{-16}$ |

*Table 1.* Standard numerical precision formats. A binary floating number is represented by a sign, an exponent and a significant. The exponent determines the dynamic range of representable values, while the significant governs the resolution, how many numbers be represented within the range. Due to the larger exponent of `bf16` it can represent numbers in a larger range, although this comes at the cost of numerical precision, which is an order of magnitude lower than `f16`. See (IEEE, 2019) and (Google Brain, 2019) for more detailed descriptions.

## B. Mixing times and spectral properties of the Markov kernel

A Markov chain with transition kernel $P$ on a finite state space $\mathcal{X}$ is irreducible if

$$\forall x, y \in \mathcal{X} \; \exists t \in \mathbb{N} \text{ s.t. } P^t(x, y) > 0.$$

The period of the chain is given by

$$d(x) := \gcd\{ t \geq 1 : P^t(x, x) > 0 \}.$$

The chain is aperiodic if $d(x) = 1$ (Levin & Peres, 2017). A central result in Markov chain theory states that if the chain is irreducible and aperiodic, there exist constants $\lambda \in (0, 1)$ and $C > 0$ such that

$$\max_{x \in \mathcal{X}} \left\| P^t(x, \cdot) - \pi \right\|_{\text{TV}} \leq C \lambda^t, \quad t \in \mathbb{N}. \tag{18}$$

This inequality expresses uniform geometric ergodicity, i.e., the distribution of the chain converges exponentially fast to its stationary distribution $\pi$, uniformly over all initial states (Meyn & Tweedie, 1993; Douc et al., 2018).

The uniform geometric ergodicity bound Eq. 18 naturally leads to the definition of the mixing time. For a prescribed accuracy $\varepsilon \in (0, 1)$, we define

$$t_{\text{mix}}(\varepsilon) := \min\left\{ t \geq 0 : \max_{x \in \mathcal{X}} \| P^t(x, \cdot) - \pi \|_{\text{TV}} \leq \varepsilon \right\}.$$

By Eq. 18, the mixing time is finite and grows at most logarithmically in the desired accuracy $\varepsilon^{-1}$.

Because $\mathcal{X}$ is finite and the MH chain is reversible with respect to $\pi$, the convergence behavior of $P$ admits a precise spectral characterization. In particular, the Markov operator $P$ is self-adjoint on $\ell^2(\pi)$ and has an orthonormal eigenbasis with real eigenvalues

$$1 = \lambda_1 > \lambda_2 \geq \cdots \geq \lambda_{|\mathcal{X}|} > -1.$$

The $\ell^2(\pi)$ contraction on centered functions is governed by

$$\rho := \max\{|\lambda_2|,\ |\lambda_{|\mathcal{X}|}|\} \in (0,1),$$

in the sense that $\|P^t f\|_\pi \leq \rho^t \|f\|_\pi$ when $\mathbb{E}[f(x)]_{x \sim \pi} = 0$. A standard $\ell^2$-to-total-variation comparison then yields

$$\max_{x \in \mathcal{X}} \|P^t(x, \cdot) - \pi\|_{\mathrm{TV}} \leq \frac{1}{2}\sqrt{\pi_{\min}^{-1} - 1}\,\rho^t. \tag{19}$$

where $\pi_{\min} := \min_{x \in \mathcal{X}} \pi(x)$.

Comparing Eq. 19 with Eq. 18, we see that the geometric convergence rate may be taken as $\lambda = \rho$. Defining the absolute spectral gap by

$$\gamma := 1 - \rho,$$

its inverse

$$t_{\mathrm{rel}} := \gamma^{-1}$$

is known as the relaxation time of the Markov chain (Levin & Peres, 2017). Combining Eq. 19 with the definition of the mixing time yields the quantitative bound

$$t_{\mathrm{mix}}(\varepsilon) \leq t_{\mathrm{rel}}\Big(\log\frac{1}{\varepsilon} + \frac{1}{2}\log\frac{1}{\pi_{\min}}\Big), \qquad \varepsilon \in (0,1).$$

Thus, the existence of a strictly positive spectral gap implies uniform geometric ergodicity, and up to logarithmic factors, the relaxation time $t_{\mathrm{rel}}$ controls the mixing time of the chain.

While spectral quantities such as $\gamma$ provide a sharp description of convergence in the reversible setting, they are often difficult to compute explicitly. In practice, it is therefore useful to bound the spectral gap from below using stronger, more readily verifiable conditions on the Markov kernel such as a Doeblin-type minorization (Douc et al., 2018). Assume that $\exists \varepsilon \in (0,1)$ and a probability measure $\nu$ on $\mathcal{X}$ such that

$$P(x, \cdot) \ \geq \ \varepsilon\,\nu(\cdot) \qquad \text{for all } x \in \mathcal{X},$$

then

$$\|\mu P - \mu' P\|_{\mathrm{TV}} \ \leq \ r\,\|\mu - \mu'\|_{\mathrm{TV}} \tag{20}$$

for all probability measures $\mu$ and $\mu'$. Hence $P$ is a strict contraction in TV with constant $r := 1 - \varepsilon$.

On a finite state space, the TV contractivity in Eq. 20 implies that all nontrivial eigenvalues of $P$ have modulus at most $1 - \varepsilon$, and hence that $P$ admits a strictly positive absolute spectral gap $\gamma \geq \varepsilon$. In this way, Doeblin's condition provides an explicit lower bound on the absolute spectral gap without requiring direct spectral analysis.

## C. Proofs

For convenience, we restate the Lemmas and Theorems in this section.

**Lemma 3.1.** *Let $\pi$ and $\tilde{\pi}$ be probability measures on $\mathcal{X}$. Let $P$ and $\tilde{P}$ be Markov kernels with invariant distributions $\pi$ and $\tilde{\pi}$, respectively. Assume that $P$ is a strict contraction in TV with constant $r \in (0,1)$ in the sense of Eq. 11. Then*

$$\|\tilde{\pi} - \pi\|_{\mathrm{TV}} \leq \left\|\tilde{\pi}(\tilde{P} - P)\right\|_{\mathrm{TV}} \frac{1}{(1-r)}.$$

*Proof of Lemma 3.1.* Using the invariance relations $\pi P = \pi$ and $\tilde{\pi}\tilde{P} = \tilde{\pi}$, we write

$$\tilde{\pi} - \pi = \tilde{\pi}\tilde{P} - \pi P = \tilde{\pi}(\tilde{P} - P) + (\tilde{\pi} - \pi)P.$$

Taking total variation norms and applying the triangle inequality yields

$$\|\tilde{\pi} - \pi\|_{\mathrm{TV}} \leq \left\|\tilde{\pi}(\tilde{P} - P)\right\|_{\mathrm{TV}} + \|(\tilde{\pi} - \pi)P\|_{\mathrm{TV}}.$$

Since $P$ is a strict contraction in total variation with constant $r \in (0, 1)$, the second term is bounded by $r\|\tilde{\pi} - \pi\|_{\mathrm{TV}}$. Rearranging yields

$$\|\tilde{\pi} - \pi\|_{\mathrm{TV}} \leq \frac{1}{1 - r}\left\|\tilde{\pi}(\tilde{P} - P)\right\|_{\mathrm{TV}},$$

which completes the proof. $\qquad\square$

**Lemma 3.2.** *Let $\mathcal{X}$ be a finite state space and let $P$ and $\tilde{P}$ be MH kernels on $\mathcal{X}$ constructed with the same proposal distribution $q(y \mid x)$ and acceptance probabilities*

$$\alpha(x, y) = \min\{1, s(x, y)\}, \quad \text{and}$$
$$\tilde{\alpha}(x, y) = \min\{1, s(x, y)e^{\varepsilon(x,y)}\}.$$

*Then, for any probability measure $\tilde{\pi}$ on $\mathcal{X}$,*

$$\left\|\tilde{\pi}(\tilde{P} - P)\right\|_{\mathrm{TV}} \leq 1 - \sum_{y \in \mathcal{X}} \mathbb{E}_{x \sim \tilde{\pi}}\left[q(y \mid x)e^{-|\varepsilon(x,y)|}\right].$$

*Proof of Lemma 3.2.* Assume that $P$ and $\tilde{P}$ are Metropolis–Hastings transition kernels on the finite state space $\mathcal{X}$, constructed from a common proposal distribution $q(\cdot \mid \cdot)$ and acceptance probabilities

$$\alpha(x, y) = \min\{1, s(x, y)\}, \qquad \tilde{\alpha}(x, y) = \min\{1, s(x, y)e^{\varepsilon(x,y)}\}.$$

For $y \neq x$,

$$P(x, y) = q(y \mid x)\alpha(x, y), \qquad \tilde{P}(x, y) = q(y \mid x)\tilde{\alpha}(x, y),$$

and the diagonal entries satisfy

$$P(x, x) = 1 - \sum_{z \neq x} q(z \mid x)\alpha(x, z), \qquad \tilde{P}(x, x) = 1 - \sum_{z \neq x} q(z \mid x)\tilde{\alpha}(x, z).$$

Define $\Delta\alpha(x, y) := \tilde{\alpha}(x, y) - \alpha(x, y)$.

**Step 1: Reduction to a row-wise kernel difference.** Let $\eta := \tilde{\pi}(\tilde{P} - P)$, so that

$$\eta(y) = \sum_{x \in \mathcal{X}} \tilde{\pi}(x)\big(\tilde{P}(x, y) - P(x, y)\big) = \mathbb{E}_{x \sim \tilde{\pi}}\big[\tilde{P}(x, y) - P(x, y)\big].$$

Since $\mathcal{X}$ is finite,

$$\|\eta\|_{\mathrm{TV}} = \frac{1}{2}\sum_{y \in \mathcal{X}} |\eta(y)|.$$

By the triangle inequality,

$$\|\tilde{\pi}(\tilde{P} - P)\|_{\mathrm{TV}} = \frac{1}{2}\sum_{y \in \mathcal{X}}\left|\mathbb{E}_{x \sim \tilde{\pi}}\big[\tilde{P}(x, y) - P(x, y)\big]\right| \leq \frac{1}{2}\mathbb{E}_{x \sim \tilde{\pi}}\left[\sum_{y \in \mathcal{X}}|\tilde{P}(x, y) - P(x, y)|\right]. \tag{21}$$

**Step 2: Bounding the row-wise difference.** Fix $x \in \mathcal{X}$. For $y \neq x$,

$$|\tilde{P}(x,y) - P(x,y)| = q(y \mid x) |\Delta\alpha(x,y)|.$$

For the diagonal term,

$$\tilde{P}(x,x) - P(x,x) = -\sum_{z \neq x} q(z \mid x) \Delta\alpha(x,z),$$

and hence

$$|\tilde{P}(x,x) - P(x,x)| \leq \sum_{z \neq x} q(z \mid x) |\Delta\alpha(x,z)|.$$

Therefore, the row-wise $\ell^1$-distance between the two kernels starting from $x$ is given by:

$$\sum_{y \in \mathcal{X}} |\tilde{P}(x,y) - P(x,y)| \leq \sum_{z \neq x} q(z \mid x) |\Delta\alpha(x,z)| + \sum_{y \neq x} q(y \mid x) |\Delta\alpha(x,y)| = 2 \sum_{y \neq x} q(y \mid x) |\Delta\alpha(x,y)|.$$

Substituting into Eq. 21 yields

$$\|\tilde{\pi}(\tilde{P} - P)\|_{\text{TV}} \leq \mathbb{E}_{x \sim \tilde{\pi}} \left[ \sum_{y \neq x} q(y \mid x) |\Delta\alpha(x,y)| \right]. \tag{22}$$

**Step 3: Bounding the acceptance error.** Fix $x \neq y$ and write $s = s(x,y) > 0$ and $\varepsilon = \varepsilon(x,y)$. We claim that

$$|\Delta\alpha(x,y)| = \left| \min\{1, se^\varepsilon\} - \min\{1, s\} \right| \leq 1 - e^{-|\varepsilon|}. \tag{23}$$

Define

$$a := \min\{1, s\}, \qquad b := \min\{1, se^\varepsilon\},$$

so that $|\Delta\alpha(x,y)| = |b - a|$.

It suffices to consider $\varepsilon \geq 0$, since the case $\varepsilon < 0$ follows by symmetry after replacing $\varepsilon$ with $|\varepsilon|$. Assume $\varepsilon \geq 0$.

If $s \geq 1$, then $a = 1$ and $se^\varepsilon \geq s \geq 1$, hence $b = 1$ and $|b - a| = 0 \leq 1 - e^{-\varepsilon}$.

If $s < 1$, then $a = s$. If $se^\varepsilon \leq 1$ (equivalently $s \leq e^{-\varepsilon}$), then $b = se^\varepsilon$ and

$$|b - a| = s(e^\varepsilon - 1) \leq e^{-\varepsilon}(e^\varepsilon - 1) = 1 - e^{-\varepsilon}.$$

If instead $se^\varepsilon > 1$ (equivalently $s > e^{-\varepsilon}$), then $b = 1$ and

$$|b - a| = 1 - s < 1 - e^{-\varepsilon}.$$

Thus $|\Delta\alpha(x,y)| \leq 1 - e^{-|\varepsilon(x,y)|}$ in all cases.

**Step 4: Conclusion.** Combining Eq. 22 with the bound from Eq. 23,

$$\|\tilde{\pi}(\tilde{P} - P)\|_{\text{TV}} \leq \mathbb{E}_{x \sim \tilde{\pi}} \left[ \sum_{y \neq x} q(y \mid x)\left(1 - e^{-|\varepsilon(x,y)|}\right) \right] \leq \mathbb{E}_{x \sim \tilde{\pi}} \left[ \sum_{y \in \mathcal{X}} q(y \mid x)\left(1 - e^{-|\varepsilon(x,y)|}\right) \right],$$

where the $y = x$ term may be added without effect. Since $\sum_{y \in \mathcal{X}} q(y \mid x) = 1$, we obtain

$$\|\tilde{\pi}(\tilde{P} - P)\|_{\text{TV}} \leq 1 - \sum_{y \in \mathcal{X}} \mathbb{E}_{x \sim \tilde{\pi}} \left[ q(y \mid x) e^{-|\varepsilon(x,y)|} \right],$$

which completes the proof. $\qquad\qquad\square$

**Theorem 3.3.** *Let $\pi$ and $\tilde{\pi}$ be probability measures on the discrete space $\mathcal{X} = \{0,1\}^n$ related by a function $\delta : \mathcal{X} \to \mathbb{R}$ such that, for all $x \in \mathcal{X}$,*

$$\log \tilde{\pi}(x) = \log \pi(x) + \delta(x).$$

*Let $P$ and $\tilde{P}$ denote the MH Markov chains targeting $\pi$ and $\tilde{\pi}$, respectively, constructed using the proposal distribution:*

$$q(y \mid x) = \begin{cases} \frac{1}{n}, & \text{if } y \in \{y_i(x)\}_{i=1}^n, \\ 0, & \text{otherwise,} \end{cases} \tag{24}$$

*where $y_i(x)$ is obtained from $x$ by flipping its $i$th bit. Assume that $P$ is a strict contraction in total variation with constant $r \in (0,1)$ in the sense of Eq. 11. Assume further that, under the joint draw $X \sim \tilde{\pi}$ and $Y \sim q(\cdot \mid X)$, the log-density increment*

$$\varepsilon(X,Y) := \delta(Y) - \delta(X)$$

*is Gaussian with mean $\mu$ and variance $2\sigma^2$. Then,*

$$\|\tilde{\pi} - \pi\|_{\mathrm{TV}} \leq \frac{1}{1-r}\left(1 - \frac{e^{\sigma^2}}{2}\left[e^{-\mu}\,\mathrm{erfc}\left(\sigma - \frac{\mu}{2\sigma}\right) + e^{\mu}\,\mathrm{erfc}\left(\sigma + \frac{\mu}{2\sigma}\right)\right]\right).$$

*Proof of Theorem 3.3.* By Theorem 3.1, it suffices to bound $\|\tilde{\pi}(\tilde{P} - P)\|_{\mathrm{TV}}$:

$$\|\tilde{\pi} - \pi\|_{\mathrm{TV}} \leq \frac{1}{1-r}\,\|\tilde{\pi}(\tilde{P} - P)\|_{\mathrm{TV}}.$$

By Theorem 3.2, since $P$ and $\tilde{P}$ are MH kernels constructed with the same proposal $q$ and perturbed acceptance probabilities, we have

$$\|\tilde{\pi}(\tilde{P} - P)\|_{\mathrm{TV}} \leq 1 - \sum_{y \in \mathcal{X}} \mathbb{E}_{x \sim \tilde{\pi}}\left[q(y \mid x)e^{-|\varepsilon(x,y)|}\right]. \tag{25}$$

For the local proposal $q(y \mid x)$ in Eq. 24, Eq. 25 becomes

$$\|\tilde{\pi}(\tilde{P} - P)\|_{\mathrm{TV}} \leq 1 - \frac{1}{n}\sum_{i=1}^n \mathbb{E}_{x \sim \tilde{\pi}}\left[e^{-|\varepsilon(x,y_i(x))|}\right].$$

Equivalently, if $X \sim \tilde{\pi}$ and $Y \sim q(\cdot \mid X)$, then

$$\frac{1}{n}\sum_{i=1}^n \mathbb{E}_{x \sim \tilde{\pi}}\left[e^{-|\varepsilon(x,y_i(x))|}\right] = \mathbb{E}\left[e^{-|\varepsilon(X,Y)|}\right],$$

and hence

$$\|\tilde{\pi}(\tilde{P} - P)\|_{\mathrm{TV}} \leq 1 - \mathbb{E}\left[e^{-|\varepsilon(X,Y)|}\right].$$

Under the assumptions of the theorem, if $X \sim \tilde{\pi}$ and $Y \sim q(\cdot \mid X)$, then the increment $\varepsilon(X,Y) = \delta(Y) - \delta(X)$ is Gaussian with mean $\mu$ and variance $2\sigma^2$. Therefore,

$$\mathbb{E}\left[e^{-|\varepsilon(X,Y)|}\right] = \mathbb{E}\left[e^{-|\varepsilon|}\right]_{\varepsilon \sim \mathcal{N}(\mu, 2\sigma^2)}.$$

We compute this expectation explicitly:

$$\begin{aligned}
\mathbb{E}\left[e^{-|\varepsilon|}\right] &= \frac{1}{\sqrt{4\pi\sigma^2}} \int_{-\infty}^{\infty} e^{-|\varepsilon|} \exp\left(-\frac{(\varepsilon - \mu)^2}{4\sigma^2}\right) d\varepsilon \\
&= \frac{1}{\sqrt{4\pi\sigma^2}} \int_0^{\infty} e^{-\varepsilon}\left(\exp\left(-\frac{(\varepsilon - \mu)^2}{4\sigma^2}\right) + \exp\left(-\frac{(\varepsilon + \mu)^2}{4\sigma^2}\right)\right) d\varepsilon.
\end{aligned}$$

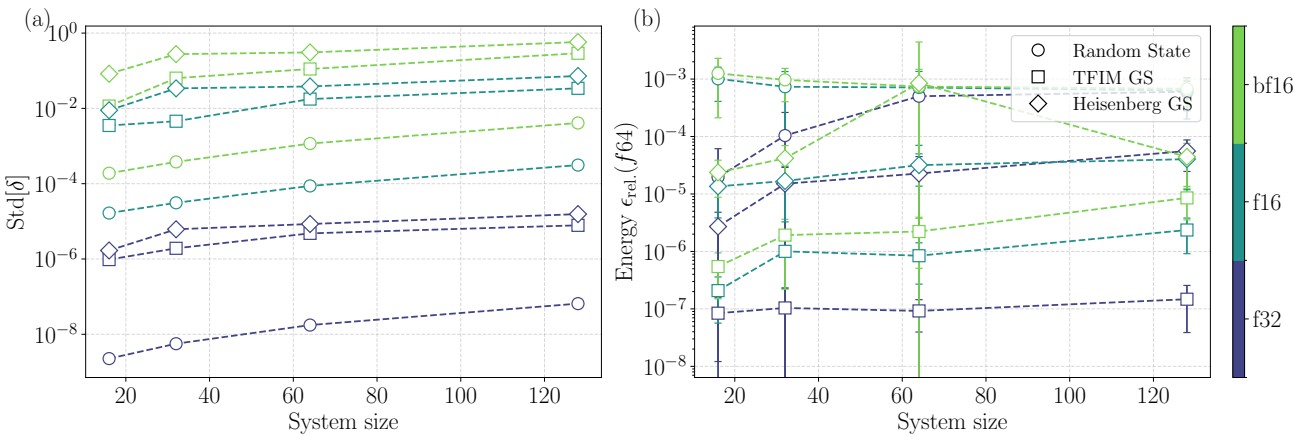

*Figure 6.* Sampling in reduced numerical precision is investigated for three one-dimensional wavefunctions: the TFIM at $h/J = 0.5$, the Heisenberg model with coupling $J = 1$, and a randomly initialized state, corresponding to a flat probability distribution. All the states are parametrized by a RBM with parameter density set to one. Panel (a) shows the standard deviation of the error term $\delta$ as a function of system size. Panel (b) reports the energies estimated via Monte Carlo sampling at different numerical precisions and compares them to the reference values obtained using double-precision arithmetic. For the randomly initialized state the TFIM Hamiltonian is used to compute the energy. The error bars indicate the MC sampling error.

For $a \in \mathbb{R}$, define

$$I(a) := \int_0^\infty e^{-\varepsilon} \exp\left(-\frac{(\varepsilon - a)^2}{4\sigma^2}\right) d\varepsilon.$$

Completing the square gives

$$-\varepsilon - \frac{(\varepsilon - a)^2}{4\sigma^2} = -\frac{(\varepsilon + 2\sigma^2 - a)^2}{4\sigma^2} + \sigma^2 - a,$$

so

$$I(a) = e^{\sigma^2 - a} \int_0^\infty \exp\left(-\frac{(\varepsilon + 2\sigma^2 - a)^2}{4\sigma^2}\right) d\varepsilon.$$

With the substitution $z = (\varepsilon + 2\sigma^2 - a)/(2\sigma)$, we obtain

$$I(a) = e^{\sigma^2 - a} 2\sigma \int_{\sigma - \frac{a}{2\sigma}}^\infty e^{-z^2} dz = e^{\sigma^2 - a} \sigma\sqrt{\pi} \operatorname{erfc}\left(\sigma - \frac{a}{2\sigma}\right),$$

using $\int_b^\infty e^{-z^2} dz = \frac{\sqrt{\pi}}{2} \operatorname{erfc}(b)$. Therefore,

$$\mathbb{E}\left[e^{-|\varepsilon|}\right]_{\varepsilon \sim \mathcal{N}(\mu, 2\sigma^2)} = \frac{1}{\sqrt{4\pi\sigma^2}}\big(I(\mu) + I(-\mu)\big)$$

$$= \frac{e^{\sigma^2}}{2}\left[e^{-\mu} \operatorname{erfc}\left(\sigma - \frac{\mu}{2\sigma}\right) + e^{\mu} \operatorname{erfc}\left(\sigma + \frac{\mu}{2\sigma}\right)\right].$$

Substituting into the bound from Theorem 3.2 yields

$$\|\tilde{\pi}(\tilde{P} - P)\|_{\mathrm{TV}} \le 1 - \frac{e^{\sigma^2}}{2}\left[e^{-\mu} \operatorname{erfc}\left(\sigma - \frac{\mu}{2\sigma}\right) + e^{\mu} \operatorname{erfc}\left(\sigma + \frac{\mu}{2\sigma}\right)\right].$$

Finally, applying Theorem 3.1 gives

$$\|\tilde{\pi} - \pi\|_{\mathrm{TV}} \le \frac{1}{1 - r}\left(1 - \frac{e^{\sigma^2}}{2}\left[e^{-\mu} \operatorname{erfc}\left(\sigma - \frac{\mu}{2\sigma}\right) + e^{\mu} \operatorname{erfc}\left(\sigma + \frac{\mu}{2\sigma}\right)\right]\right),$$

which is exactly Eq. 14. $\qquad\square$

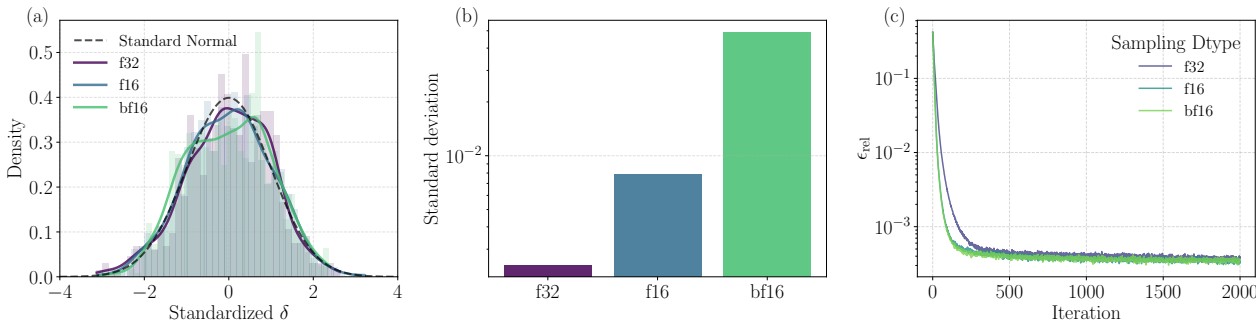

*Figure 7.* Panel (a) shows the distribution of the error term $\delta$ for the ground state of the Heisenberg model with coupling $J = 1$ on a chain of length 12, computed using different numerical precisions. The distributions have been standardized to allow a direct comparison of their shapes. Panel (b) presents the corresponding standard deviations of these distributions. Panel (c) reports the relative error of the loss functions as a function of training step, obtained using our mixed-precision optimization scheme for the same model with 64 spins and using as architecture an RBM with parameter density set to one. The relative error is evaluated by comparing the estimated energies with reference values obtained via density matrix renormalization group (DMRG), which provides near-exact ground-state energies in one dimension.

## D. System Size Scaling

To investigate how the error term $\delta$ scales with system size, we perform the following numerical experiment. We train a set of RBMs while varying both the number of spins in the one-dimensional chain and the underlying physical model. In particular, we optimize the networks using two different Hamiltonians: the TFIM with $h/J = 0.5$ and the Heisenberg model with coupling $J = 1$. After training, all NQS are downcast from double precision to several reduced-precision data types, and the standard deviation of the error term $\delta$ is estimated for each case. The same analysis is carried out for a randomly initialized network, corresponding to a flat probability distribution.

As shown in panel (a) of Fig. 6, the standard deviation of $\delta$ exhibits a mild increase with system size. This behavior can be attributed to the growing number of variational parameters and, consequently, to the larger number of arithmetic operations performed during each forward pass of the network. The increased operation count leads to a greater accumulation of numerical errors at the model output. Nevertheless, the error remains well controlled and stays below unity for all system sizes and models considered.

In addition, we use the networks at different numerical precisions to estimate the energy of the corresponding Hamiltonians via Monte Carlo sampling. For the randomly initialized state, we instead compute the energy of the TFIM. As illustrated in panel (b) of Fig. 6, the relative energy error remains approximately constant as a function of system size. Furthermore, a clear dependence on the structure of the underlying probability distribution is observed. The TFIM ground state, which is the most strongly peaked distribution, exhibits the smallest relative error. This is followed by the Heisenberg ground state, which remains relatively peaked but is broader due to enhanced quantum fluctuations. Finally, the flat distribution yields the largest relative error. These observations are consistent with the theory derived in the main text.

## E. The Heisenberg Model

The (spin-1/2) Heisenberg model is defined by the Hamiltonian

$$H = J \sum_{\langle i,j \rangle} \sigma_i \cdot \sigma_j, \tag{26}$$

where $\sigma_i = (\sigma_i^x, \sigma_i^y, \sigma_i^z)^T$ and $\langle \cdot \rangle$ indicates the two nearest neighbors. Here we focus on the antiferromagnetic regime, $J = 1$, in which neighboring spins energetically prefer to align antiparallel. In contrast to the transverse-field Ising model, the Heisenberg interaction is fully rotationally invariant in spin space and introduces quantum fluctuations intrinsically through the non-commuting spin components. The competition between different spin orientations on each bond prevents the simultaneous minimization of all interaction terms, giving rise to strong quantum correlations.

In this section, we repeat the same experiments for the Heisenberg model that were carried out in the main text for the TFIM.

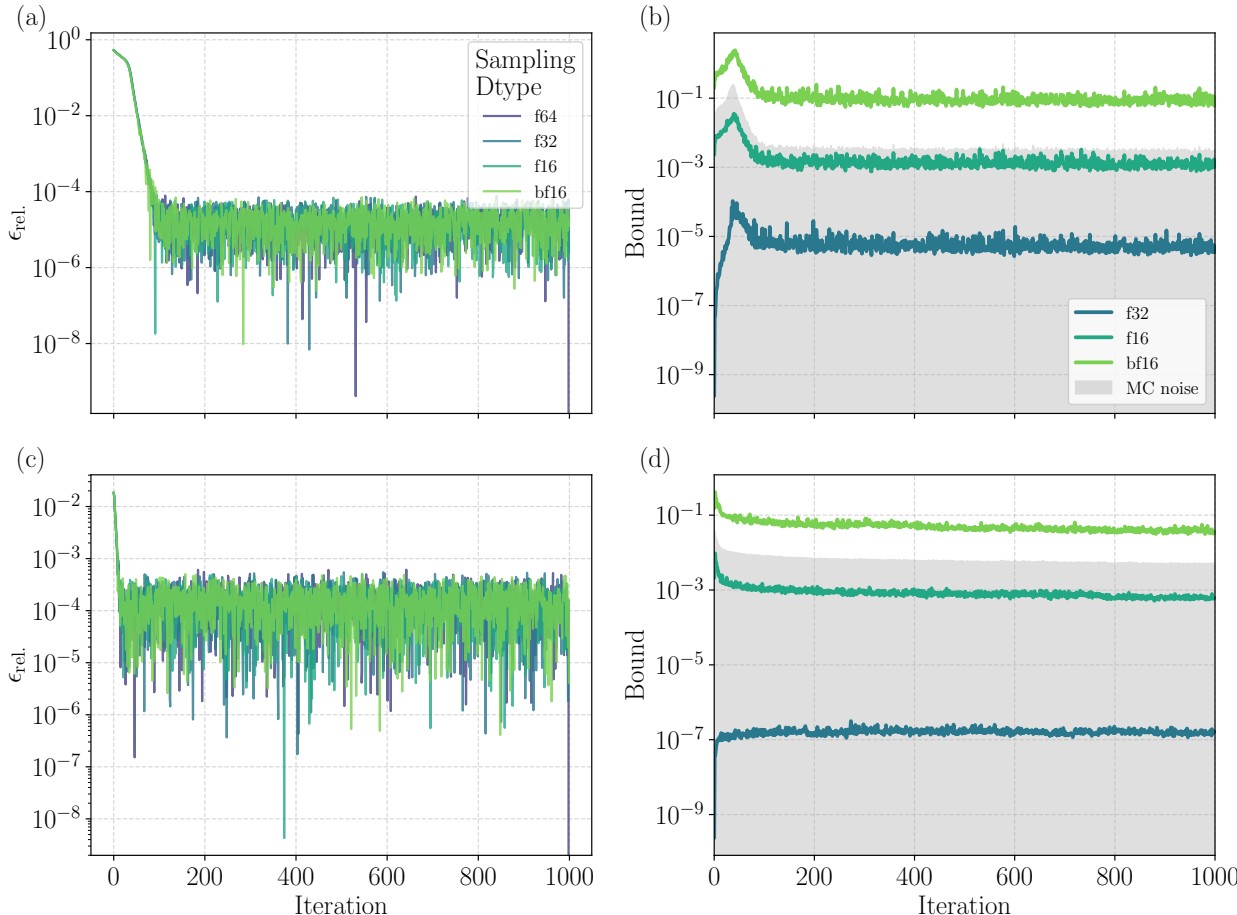

*Figure 8.* Panels (a, b) show the relative energy error with respect to the double-precision reference solution as a function of training step. Optimization is performed using the mixed-precision scheme introduced in this work, with sampling carried out in reduced precision. We use a ResCNN with kernel size $(3, 3)$, four residual blocks, and 16 filters. Panels (c, d) report, for the same optimizations, the minimum between the KL bound and the bound in Eq. 14, evaluated on the absolute difference between gradients from perturbed and unperturbed distributions. Shaded regions indicate the Monte Carlo confidence interval, given by $3\sqrt{2}\sqrt{\mathrm{Var}[\nabla E_\theta(x)]/N_{\mathrm{samples}}}$. The optimization are carried out for the two-dimensional TFIM with linear system size $L = 10$ $(N = L^2)$, for (a, c) $h/J = 1$ and (b, d) $h/J = 5$

Importantly, because the total magnetization is conserved in the Heisenberg model, we used a different sampling strategy compared to the TFIM. In the TFIM, a new configuration is generated by randomly selecting a single spin from the previous configuration and flipping it. In contrast, for the Heisenberg model, two spins are randomly chosen and their positions are swapped, ensuring that the total magnetization remains unchanged.

The results are shown in Fig. 7. From panels (a) and (b), we observe that the distribution of $\delta$ is approximately Gaussian and that its standard deviation is small. In panel (c), we apply our mixed-precision optimization scheme, performing the sampling in low precision, for a chain of 64 spins while optimizing the expectation value of the Heisenberg Hamiltonian. In this case as well, the loss function as a function of training steps is nearly identical across different data types, demonstrating that the mixed-precision approach does not compromise the accuracy of the optimization.

## F. Optimizations for the Two Phases of the TFIM

In this section, we repeat the experiment presented in the main text for the two distinct phases of the TFIM, away from criticality. Specifically, we optimize a neural quantum state based on a ResCNN architecture while sampling in different

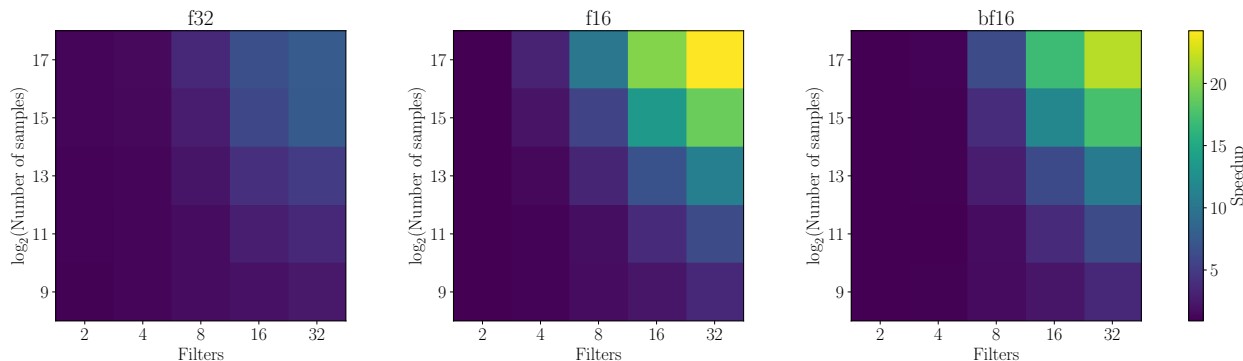

*Figure 9.* Speedup obtained by sampling from ResCNN with different data types. The results are shown as a function of both the number of samples $N_s$, with the number of Markov's chains set to $N_s/4$, and the number of filters. The number of residual blocks and kernel size are fixed to $4$ and $(3, 3)$ respectively, while the system size is fixed to $4 \times 4$.

numerical precisions, using the two-dimensional TFIM as the Hamiltonian. We consider both the antiferromagnetic phase at $h/J = 1$ and the paramagnetic phase at $h/J = 5$. The results, shown in Fig. 8, provide further evidence that reduced-precision sampling does not affect the optimization dynamics, as the Monte Carlo noise consistently dominates the error introduced by low-precision arithmetic.

## G. Restricted Boltzmann Machine

The restricted Boltzmann machine (RBM) (Carleo & Troyer, 2017; Melko et al., 2019) is a shallow neural network architecture. Given an input configuration $x$, the output of the RBM can be expressed as:

$$\log \psi_\theta(x) = a \cdot x + \sum_{i=1}^{\alpha \cdot N} \log \cosh\left(\left(Wx + b\right)_i\right) \tag{27}$$

where $a \in \mathbb{C}^N$ and $b \in \mathbb{C}^{\alpha N}$ are visible and hidden biases respectively, $W \in \mathbb{C}^{\alpha N \times N}$ is a weight matrix, and $\alpha$ is a scaling factor of the input dimension. As a consequence the learnable parameters consist of $\theta = \{a, b, W\}$.

## H. ResCNN

In our work we also considered a convolutional residual neural network (ResCNN) as implemented in (Barton et al., 2026). Given a two dimensional input configuration $x$, the model first applies an initial convolutional layer embedding:

$$h^{(0)} = \text{Conv}(x), \tag{28}$$

where Conv denotes a linear convolution with learnable kernels. Then $N_{\text{res}}$ residual blocks are sequentially applied as follows:

$$h^{(\ell+1)} = h^{(\ell)} + \text{Conv}\left[\phi\left(\text{Conv}\left(\phi\left(\text{LN}\left(h^{(\ell)}\right)\right)\right)\right)\right], \qquad \ell = 0, \ldots, N_{\text{res}} - 1, \tag{29}$$

where $\text{LN}(\cdot)$ denotes layer normalization and $\phi(\cdot)$ is a pointwise nonlinearity (GELU in our implementation). After the residual stack, a final layer normalization is applied and the scalar network output is obtained by summing over all spatial and channel indices:

$$\log \psi_\theta(x) = \sum_{i,j,c} \text{LN}\left(h^{(N_{\text{res}})}\right)_{ijc}. \tag{30}$$

In this section, we analyze the sampling speedup achieved using a ResCNN architecture with a kernel size of $(3, 3)$ and $4$ residual blocks. The results, shown in Fig. 9, confirm the trends already discussed in the main text. Increasing both the number of samples and the number of parameters in the architecture leads to a larger speedup, primarily due to the more efficient utilization of GPU memory. Notably, we observe a substantial speedup when moving from double to single precision. This can be attributed to the fact that convolution operations are highly optimized in single precision. By contrast, the performance difference between single and half precision remains comparable to that observed for linear layers.

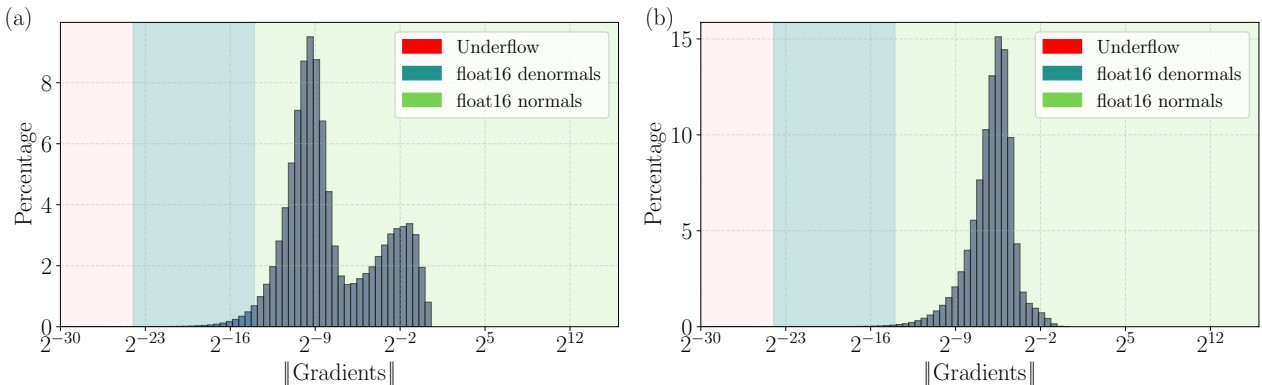

*Figure 10.* Dynamic range of gradients from an optimization performed in double precision for both the TFIM (panel (a)) and Heisenberg model (panel (b)) on a chain of 64 spins. The figures indicate the normal, denormal, and underflow regions corresponding to float16 representation.

## I. Effects on Other Parts of VMC

While the main text focused on the effects of using low-precision data types on the sampling process, in the following we discuss their impact on the other components of the VMC algorithm. In particular, we examine how performing the gradient computation and the preconditioning step in low precision influences the results.

### I.1. Gradients in Low Precision

The loss function $E_\theta$ is estimated by calculating the average $E_\theta = \mathbb{E}[E_\theta^l(x)]$ of the "local energies", as defined in Eq. 3.

The gradient of the energy is given by the force vector

$$F := \nabla_\theta E_\theta = \mathbb{E}\left[O^*(x)E_\theta^l(x)\right] - \mathbb{E}\left[O^*(x)\right]\mathbb{E}\left[E_\theta^l(x)\right] \tag{31}$$

where $O_n^*(x) = \partial_{\theta_n} \log \psi_\theta^*(x)$.

The forces $F$ depend on the derivatives of the log-amplitude $O^*(x)$. In low-precision data formats we have less resolution and so gradients that are too small may be flushed to zero which can inhibit training convergence. Large gradients on the other hand, may overflow to infinity, leading to unstable training and potential divergence.

To mitigate these issues, several strategies have been proposed in the context of mixed-precision training (Micikevicius et al., 2018). One effective technique is to perform numerically sensitive operations, such as reductions (e.g., summations over large vectors) and dot products, in higher precision before casting the result back to a lower-precision format. This prevents the systematic accumulation of rounding errors without significantly impacting performance. Importantly, this compromise preserves most of the computational speedup, since the dominant cost in neural network training lies in large-scale dense matrix multiplications, which can still be efficiently executed in reduced precision.

Complementary to these algorithmic safeguards, it is crucial to analyze the expected range of gradient values for a given problem. If gradients systematically fall into the subnormal regime, or worse outside the representable range, training becomes unreliable. A common remedy is loss scaling, where the loss function is multiplied by a constant factor during backpropagation to ensure that gradients remain within a well-represented numerical range. After backpropagation, the gradients are rescaled accordingly before being applied to the parameters.

Before training the NQS in low precision, we first performed an optimization in double precision, recording the gradients at each iteration to analyze their dynamic range throughout the optimization process. This procedure was carried out for both the one-dimensional Heisenberg and TFIM models with 64 spins. As illustrated in Fig. 10, only a small fraction of the gradients (less than 2%) enter the subnormal range, while all values remain above the threshold at which they would underflow in half precision. Importantly, this indicates that no loss rescaling is necessary to perform the optimization in low precision.

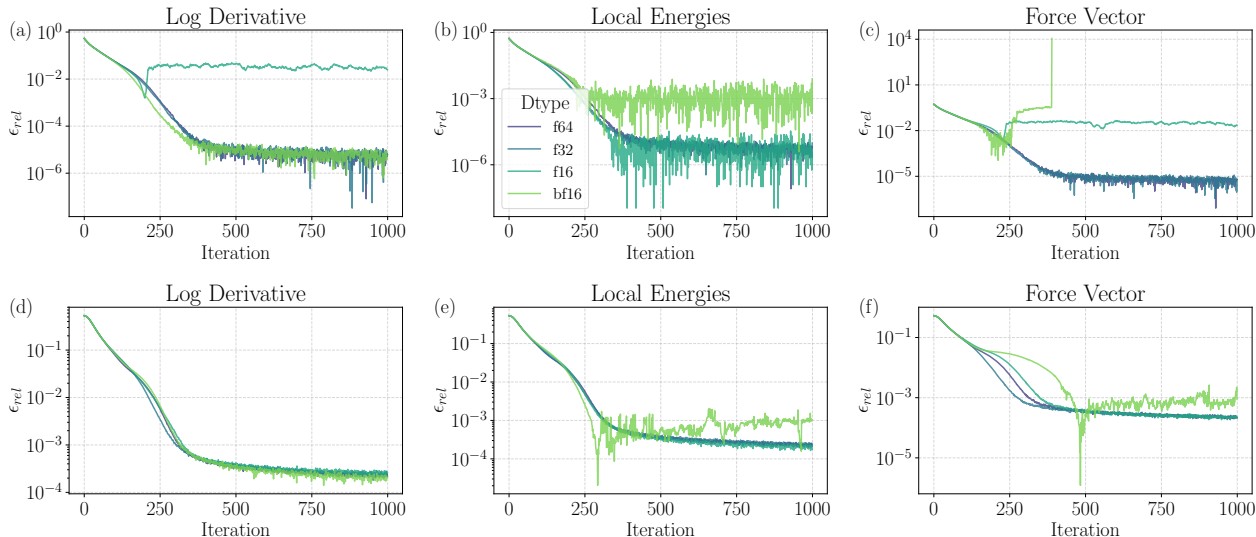

*Figure 11.* VMC optimizations in which low-precision data types are used to compute (a, d) the logarithmic derivatives $O(x)$, (b, e) the local energies $E^l(x)$, and (c, f) the full force vector. Panels (a, b, c) correspond to a weak regularization, implemented via a diagonal shift of $10^{-3}$ in the inversion of the $S$ matrix, whereas panels (d, e, f) use a stronger regularization of $10^{-1}$. All optimizations are performed using an RBM with parameter density equal to 1, applied to the TFIM on a spin chain of length 64, with $2^{17}$ Monte Carlo samples. The relative error is computed with respect to the ground-state energy obtained from DMRG.

## I.2. Stochastic Reconfiguration in Low Precision

The canonical approach for optimizing any quantum state $|\psi_\theta\rangle\langle\psi_\theta|$ is the method of Stochastic Reconfiguration (SR) (Sorella, 2005), which is the complex analogue of Natural Gradient Descent (Amari, 1998). The SR update step for the parameters $\theta$ is given by

$$g = S^{-1}F$$
$$\theta \mapsto \theta - \eta g \tag{32}$$

for a chosen learning rate $\eta$, with

$$S_{nm} = \mathbb{E}\left[O_n^*(x)O_m(x)\right] - \mathbb{E}\left[O_n^*(x)\right]\mathbb{E}\left[O_m(x)\right] \tag{33}$$

In practice a Tikhonov regularization scheme is used to shift the diagonal of the matrix $S$ by a constant $S \mapsto S + I\lambda$ to avoid numerical instabilities in the inversion.

For the inversion of the SR step in Eq. 32, there are two sources of error that affect the final parameter update. First, consider the system $S\tilde{g} = F + \delta F$ where $\delta F$ is the error in the force vector and $\tilde{g}$ is the resulting solution. We find that

$$\frac{\|\Delta g\|}{\|g\|} \leq \kappa(S)\frac{\|\delta F\|}{\|F\|}, \quad \Delta g = \tilde{g} - g, \tag{34}$$

hence errors in the force vector are amplified by the condition number $\kappa(S) = \|S^{-1}\|\|S\|$ of the $S$-matrix. On the other hand, if we look at the system $(S + \delta S)g = F$. We see that the error $\delta S$ gets amplified in a similar fashion

$$\frac{\|\Delta g\|}{\|g\|} \leq \frac{\kappa(S)\frac{\|\delta S\|}{\|S\|}}{1 - \kappa(S)\frac{\|\delta S\|}{\|S\|}}. \tag{35}$$

Hence, even if the error in computing gradients in low precision might be small, this gets amplified by the condition number of the $S$-matrix. In particular, we expect that for ill-conditioned $S$-matrices where $\kappa(S)$ is large by definition, we will be affected more by rounding errors due to low-precision. In such cases, it may be necessary to further regularize the $S$-matrix

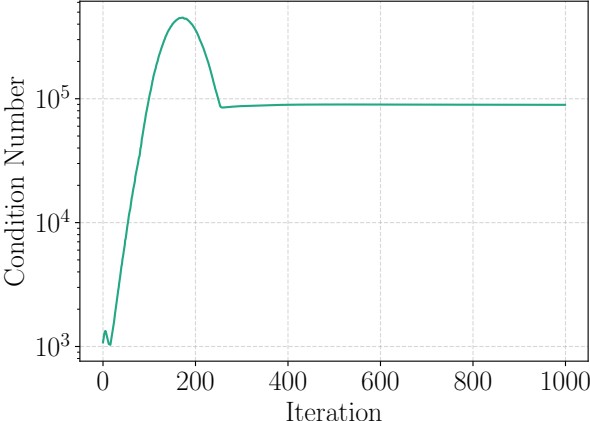

*Figure 12.* Condition number of the $S$ matrix evaluated at each training step during a VMC optimization of the TFIM on a spin chain of length 64.

using larger diagonal shifts $\lambda$, which can slow down the convergence of training and, in some instances, even degrade the final quality of the wave function. Importantly, as shown in Eq. Eq. 34, the error due to low precision can compromise the inversion of the matrix, and as a consequence the whole optimization, even though only the right hand side of the equation, the force vector, is computed in low precision, while the matrix itself is computed in high precision.

### I.3. Optimization with Gradients in Low Precision

In this section, we numerically assess the considerations discussed above. Specifically, we design three distinct optimization protocols. As evident from Eq. 31, the forces depend on two ingredients: the gradients of the logarithmic derivative $O(x)$ and the local energies $E^l(x)$. Accordingly, we consider the following tests: (1) an optimization in which only the gradients of the logarithmic derivative are computed in low precision; (2) an optimization in which only the local energies are computed in low precision; and (3) an optimization in which the entire force vector is computed in low precision. To isolate the impact of low-precision errors on the gradients, all remaining components of the VMC algorithm, namely the sampling procedure and the $S$ matrix, are evaluated in double precision.

As shown in panels (a, b, c) of Fig. 11, the optimization yields essentially identical results when single precision is used in place of double precision, whereas the errors introduced by `f16` and `bf16` are sufficient to render the optimization highly unstable. Moreover, the instability is exacerbated when the entire force vector is computed in low precision. To further investigate this behavior, we repeat the same optimization while increasing the regularization of the $S$ matrix, raising the diagonal shift from $10^{-3}$ to $10^{-1}$. As illustrated in panels (d, e, f) of Fig. 11, the optimization becomes significantly more stable, indicating that errors arising in the inversion of the $S$ matrix are the dominant source of instability. As expected, however, the increased regularization also results in a degradation of the final accuracy.

To further substantiate this interpretation, we compute the condition number of the $S$ matrix at each training step. As shown in Fig. 12, the condition number exhibits a pronounced peak during the early stages of training. Notably, the location of this peak coincides with the epoch at which the optimizations performed with different data types in Fig. 11 begin to diverge. This observation confirms that regions of parameter space characterized by a large condition number of the $S$ matrix amplify low-precision errors, ultimately leading to divergent and frequently unstable optimization trajectories.

This preliminary study highlights the challenges associated with computing gradients in low precision, even when they lie within the normal dynamic range (see Fig. 10). In particular, it shows that more sophisticated training strategies are required to ensure stability. While a more systematic analysis is deferred to future work, it is clear that stronger regularization or alternative techniques aimed at stabilizing the matrix inversion may come at the cost of reduced final accuracy or slower convergence, potentially offsetting the advantages of employing low-precision data types in this stage of VMC.

