# OpenReview forum: "Neural Quantum States in Mixed Precision"
_ICML.cc/2026/Conference — ICML 2026 regular_

### Official Review · Reviewer_33aJ · 2026-02-24

**Soundness:** 4
**Presentation:** 3
**Significance:** 3
**Originality:** 2
**Overall Recommendation:** 5
**Confidence:** 3

**Summary:**

The paper studies the impact of reduced numerical precision on the bias of expectation values, both theoretically and on the application to variational Monte Carlo. It provides analytic bounds on the bias as a function of the target density noise and compares the bounds with numerical experiments. It also showcases problems where reduced precision does not impair precision and offers considerable speed-ups.

**Compliance With Llm Reviewing Policy:**

Affirmed.

**Final Justification:**

The authors have adequately answered my questions, have added new experiments which are more convincing.

**Key Questions For Authors:**

I have some questions about Figure 1 b):

1. The regime where the bounds are tested is a small portion of the figure (essentially sigma > 1 region), because the rest is dominated by the Monte Carlo variance. An experiment that would test the bounds in a more meaningful way would be useful. If running more samples is too expensive, a smaller or simpler model could be used.
2. In the regime where the bounds are valid they are not particularly tight, so it is not clear how useful they are in practice? Also, the Avg bound doesn’t seem stronger than the KL bound? Why don’t we see the O(sigma) and O(sigma^2) scaling of the Avg bound in the limit sigma -> 0?

**Limitations:**

Yes.

**Strengths And Weaknesses:**

# Soundness
The paper is technically sound and the stronger assumptions in the theorems (Gaussian distribution of delta) are supported by numerical experiments.

# Presentation

Generally, the paper is logically organized and easy to follow. It was a pleasure to read. I appreciate that various results in the Figures are intuitively explained in the main text.
I only have a few minor comments:

I think paragraph below Equation 12 assumes epsilon > 0, because the case epsilon < 0 is symmetric. This is ok, but it would improve readability if this was explicitly stated in the text, not just in the Figure caption.
\varepsilon and \epsilon are used for two different things, the local energy and the difference of deltas. It is clear from the context which one is used, but I still think it would be clearer if a different letter was used for one of them. Local energy is only defined implicitly in Equation 2, it would be easier to read if it was defined explicitly.
Line 340 on the right: It is not clear what the “this hypothesis” refers to.
Some details about the experiments in Figure 4 would be useful. How is alpha (density of parameters) defined? How is the speed-up defined, is it wallclock time to the converged ground state? If so, how is “converged” defined?

# Significance
I view the paper as a useful reference for all future work considering reduced numerical precision. As such I believe it is significant and useful.

# Originality
The idea of using reduced precision in MCMC is not new and the approach that the authors take is also not particularly new, but they do offer new theoretical insight and the basis to assess the impact of reduced precision on the sampling results.

---

> ### Author Rebuttal · Authors · 2026-03-30
>
> **Soundness – Significance**
>
> We are pleased to hear that the reviewer considers our work a useful reference for future research on numerical precision, and we thank them for recommending our paper for acceptance.
>
> **Presentation 1**
>
> We thank the reviewer for this positive assessment and are glad they found the paper a pleasure to read.
>
> **Presentation 2**
>
> We thank the reviewer for these helpful suggestions. We will address all three points in the revised manuscript: we will explicitly state the assumption $\epsilon > 0$ below Equation 12, introduce distinct notation for the local energy and the difference of deltas, and provide an explicit definition of the local energy at Equation 2.
>
> **Presentation 3**
>
> We thank the reviewer for pointing this out. In line 340, “this hypothesis” refers to the statement in line 334: *“the variance of $\delta$ is consistently smaller than 1, indicating that MCMC should be able to converge reliably to the underlying distribution despite the reduced precision.”* We will revise the text to make this reference explicit.
>
> **Presentation 4**
>
> We thank the reviewer for raising these points. We will add the missing details in the revised manuscript.
>
> The parameter density $\alpha$ is a standard RBM quantity defined as the ratio of the number of parameters to the number of inputs (i.e., the number of spins). We will state this explicitly in the main text.
>
> Regarding the speed-up, the experiment in Figure 4 measures the average execution time of a single call to the sampling subroutine, so no notion of convergence is required. The speed-up is defined as:
> $$
> \text{Speed-up} = \frac{t_{f64}}{t_{\delta}}.
> $$
> We will clarify this in the revised manuscript.
>
> **Question 1**
>
> We thank the reviewer for this insightful suggestion. We agree that reducing Monte Carlo noise is key to testing the bounds over a wider range of $\sigma$. While the number of samples used in the original experiment is already large ($2^{18}$), we explored an alternative strategy to suppress statistical fluctuations more effectively.
>
> In particular, we designed an experiment leveraging the zero-variance principle. Instead of evaluating a generic observable, we first optimize the NQS to approximate an eigenstate of a chosen operator, and then measure that same operator. In this setting, the estimator variance is strongly reduced, allowing for a much cleaner comparison with the theoretical bounds. This setup is also relevant in practice, as energy estimation during later stages of training benefits from the same principle.
>
> Concretely, we considered a spin chain of 8 sites and optimized the NQS for the transverse-field Ising model in a regime where the ground state closely approximates an eigenstate of the total magnetization $M_x$. We then initialized the noisy RBM from the optimized parameters and measured the relative error in $\langle M_x \rangle$ as a function of $\sigma$.
>
> This setup significantly reduces Monte Carlo noise and allows us to verify that the proposed bounds remain satisfied down to $\sigma \sim 10^{-2}$, well beyond the regime visible in the original figure.
>
> Due to rebuttal constraints, we cannot include the figure here, but we report the results:
>
> - $\sigma = 4.833 \times 10^{-3}$: Bound $= 1.086 \times 10^{-2}$, $3\times$ MC error $= 2.095 \times 10^{-5}$, Relative error $= 1.159 \times 10^{-5}$
> - $\sigma = 2.637 \times 10^{-2}$: Bound $= 5.814 \times 10^{-2}$, $3\times$ MC error $= 1.136 \times 10^{-4}$, Relative error $= 6.906 \times 10^{-4}$
> - $\sigma = 1.438 \times 10^{-1}$: Bound $= 2.873 \times 10^{-1}$, $3\times$ MC error $= 6.150 \times 10^{-4}$, Relative error $= 2.149 \times 10^{-2}$
> - $\sigma = 1.833$: Bound $= 1.451$, $3\times$ MC error $= 8.645 \times 10^{-3}$, Relative error $= 9.358 \times 10^{-1}$
>
> **Question 2**
>
> We thank the reviewer for this important observation. There was indeed a small mistake in the analysis that we identified after submission.
>
> Upon closer inspection, we found that the claim regarding $\sigma^2/(1 - r)$ scaling is incorrect. A more careful analysis shows that the Avg bound also scales linearly in $\sigma$, consistent with the reviewer’s observation.
>
> We emphasize that this correction does not affect the conclusions of the paper. The value of the Avg bound lies not in improved asymptotic scaling, but in providing a more faithful description of the MCMC sampling process. By incorporating the contraction rate $r$, it explicitly connects autocorrelation and thermalization properties of the sampler to the impact of reduced precision.
>
> Regarding practical utility: the bound is inexpensive to evaluate and can serve as a diagnostic tool. As illustrated in Figure 3, one can compare the bound to Monte Carlo noise—if the latter dominates, reduced-precision errors are negligible in practice. We will make this point more explicit in the revised manuscript.

---

> > ### Author Rebuttal · Reviewer_33aJ · 2026-04-01
> >
> > The authors have answered my questions.

---

### Official Review · Reviewer_24cA · 2026-02-28

**Soundness:** 2
**Presentation:** 2
**Significance:** 2
**Originality:** 3
**Overall Recommendation:** 4
**Confidence:** 3

**Summary:**

This paper studies the effect of using reduced-precision arithmetic in Metropolis–Hastings (MH) MCMC sampling, and in particular, for variational Monte Carlo (VMC) for neural quantum states. It models the rounding error as an additive Gaussian noise for the log density function, leading to a perturbed stationary distribution. It shows analytic bounds on the bias in stationary expectations under low precision. The experiments show that sampling can often be performed in half-precision without noticeable degradation in training curves or final energies, while delivering substantial speedups.

**Compliance With Llm Reviewing Policy:**

Affirmed.

**Final Justification:**

The authors have answered my question, but the significance and technical novelty of this work are marginal.

**Key Questions For Authors:**

1. This paper claims that "the analysis of mixed-precision effects in the context of MCMC sampling employed in machine learning, or,
NQS in particular, remains underdeveloped." However, I did a quick Google search and found a paper titled "A Mixed Precision Monte Carlo Methodology for Reconfigurable Accelerator Systems" by Chow et al. I wonder whether it is a relevant work.

2. Can the contraction constant $r$ be replaced by an empirical proxy (e.g., autocorrelation or relaxation time estimates) and shown to correlate with observed robustness to mixed precision?

**Limitations:**

yes

**Strengths And Weaknesses:**

**Strengths:** The research problem studied in this paper is an important problem in sampling and optimization. It is a natural and interesting idea to adapt the use of mixed or low-precision in optimization to sampling, and particularly to neural quantum states. The numerical experiments also validate the model and demonstrate the benefits of mixed-precision.

**Weaknesses:** First, the theoretical claim about the $O(\sigma^2)$ bias is not mathematically sound. Tthe term $1-e^{\sigma^2}\mathrm{erfc}(\sigma)$ behaves linearly in $\sigma$ near 0, so the bound is $\Theta(\sigma)/(1-r)$, not $\Theta(\sigma^2)/(1-r)$. Thus, the improvement in order over the straightforward approach may not hold.

Second, the bias depends on $\frac{1}{1-r}$, which can be large in large MH problems. Furthermore, this quantity is hard to estimate in practice, so the tightness and utility of the bound in realistic NQS settings remain unclear.

Third, in Table 1, the 'f16' row, 4.88 × 10−5 should be 4.88 × 10−4

---

> ### Author Rebuttal · Authors · 2026-03-30
>
> **Strengths**
>
> We thank the reviewer for acknowledging the importance of the problem studied in our work.
>
> **Weakness 1**
>
> We thank the reviewer for this careful observation. We had also identified this issue after submission. The reviewer is correct: the term $1 - e^{\sigma^2}\mathrm{erfc}(\sigma)$ behaves as $\Theta(\sigma)$ near $\sigma = 0$, and therefore the Avg bound scales as $\Theta(\sigma)/(1 - r)$, not $\Theta(\sigma^2)/(1 - r)$ as we incorrectly claimed.
>
> We will correct the corresponding statement at the end of the theory section in the revised manuscript. Specifically, we will revise the claim regarding the asymptotic scaling of the bound as $\sigma \to 0$.
>
> We emphasize that this correction does not affect the validity of the Avg bound itself, nor the conclusions of our paper. The purpose of the Avg bound is not to improve asymptotic scaling, but to provide a more accurate description of the sampling process by incorporating the contraction rate $r$, and therefore the autocorrelation and thermalization properties of the sampler, into the analysis. We believe this makes it the more appropriate bound for understanding the role of mixed precision in VMC.
>
> **Weakness 2**
>
> Thank you for raising this point. We agree that the prefactor $1/(1 - r)$ can become large for slowly mixing Metropolis–Hastings chains, and that the exact contraction constant is generally difficult to compute.
>
> However, we view this as a strength of the analysis. When the sampler mixes rapidly, mixed-precision noise induces only a small asymptotic bias; when mixing is slow, the bound becomes weak and correctly signals that no strong robustness guarantee should be expected. The bound therefore provides both qualitative insight (slow mixing increases sensitivity to numerical errors) and quantitative guidance (fast mixing controls the impact of $\sigma$).
>
> In practice, one can use observable-based diagnostics, such as the integrated autocorrelation time of a slow observable, as a proxy for how close the chain is to the slow-mixing regime. While such diagnostics are observable-dependent and do not coincide exactly with the theorem’s $r$, they provide a practical way to assess whether the sampler operates in a regime where the bound is informative.
>
> We also note that for the systems considered in our experiments, mixing times are relatively short, and we did not encounter issues arising from this factor.
>
> **Weakness 3**
>
> We thank the reviewer for spotting this mistake. We will correct it in the final version.
>
> **Question 1**
>
> We thank the reviewer for pointing out this relevant reference. The work by Chow et al. indeed studies mixed-precision techniques in the context of Monte Carlo simulations, but its focus differs substantially from ours in both scope and methodology.
>
> In particular, Chow et al. propose a hardware-oriented mixed-precision framework for Monte Carlo simulations on FPGA-based systems, where reduced-precision computations are corrected via an auxiliary sampling procedure to eliminate finite-precision bias. In contrast, our work focuses on MCMC sampling as used in machine learning, and specifically in neural quantum states. In this setting, sampling is correlated (Markov chains rather than i.i.d.), and the Metropolis–Hastings acceptance step introduces nonlinear sensitivity to numerical errors.
>
> In the main text, we emphasize that while prior work has explored reduced-precision arithmetic in sampling algorithms, there is currently no systematic study of its effects within VMC, and in particular for NQS. Our goal is therefore to provide both intuition and practical tools to assess whether the sampling component of VMC can be reliably performed in reduced precision.
>
> We believe this is a fundamental problem for two main reasons. First, VMC simulations are computationally demanding and often require substantial GPU resources, making it crucial to identify which components can be accelerated without compromising accuracy. Second, modern hardware is increasingly optimized for lower-precision arithmetic, making it essential to understand whether VMC can operate reliably in such regimes.
>
> **Question 2**
>
> While $r$ in Theorem III.3 is a worst-case total-variation contraction constant, we can define an empirical surrogate based on a slow observable $f$:
> \[
> \rho_f(k) := \frac{\mathrm{Cov}_\pi(f(X_0), f(X_k))}{\mathrm{Var}_\pi(f)}, \quad
> \tau_{\mathrm{int}}(f) := 1 + 2 \sum_{k \geq 1} \rho_f(k).
> \]
>
> From this, one can define
> $$
> r_{\mathrm{emp}} := \frac{\tau_{\mathrm{int}}(f) - 1}{\tau_{\mathrm{int}}(f) + 1}.
> $$
>
> This provides an observable-dependent proxy for the effective geometric convergence rate $\rho$, for which $\rho \le r$. In practice, this allows us to estimate robustness to mixed precision by combining such empirical diagnostics with the noise level $\sigma$.

---

> > ### Author Rebuttal · Reviewer_24cA · 2026-04-03
> >
> > We thank the authors for the detailed response to my questions.

---

### Official Review · Reviewer_FtF9 · 2026-03-10

**Soundness:** 3
**Presentation:** 3
**Significance:** 3
**Originality:** 3
**Overall Recommendation:** 5
**Confidence:** 2

**Summary:**

This paper studies the use of mixed precision for accelerating variational monte carlo (VMC) with neural quantum states (NQS). The authors achieved accelerated speed without sacrificing accuracy. The key idea is that sampling can be performed in lower precision without significantly affecting the stationary distribution, while gradient computation and stochastic reconfiguration are more sensitive due to conditioning issues.

**Compliance With Llm Reviewing Policy:**

Affirmed.

**Final Justification:**

My initial thought was that the Gaussian assumption is strong and essential in this paper. However, as authors pointed out, it can be considered as future work and with central limit theorem and other empirical evidences. I still think that as it is ICML, more extensive experiments can be brought up, but I assume this is more theoretical paper rather than experimental. As I am not largely familiar with this domain, I remain my score as 5 with regard to the aspects mentioned.

**Key Questions For Authors:**

* How does the algorithm performs when the correction rate is modified? Is there any rule/suggestion for this rate decision?
* How does it perform when we use other distribution instead of Gaussian?

**Limitations:**

Yes the authors included impact statement of the paper and provided limitations of the paper.

**Strengths And Weaknesses:**

## Strengths

- this paper addresses a practical bottleneck in neural quantum states and variational monte carlo workflows.
- clear separation of precision sensitivity in the vmc pipeline
- practical insight

## Weaknesses

- assumption on Gaussian distribution error - Even the authors empirically shown the result, this assumption lacks theoretical insight. how can we extend to non-gaussian or adversarial error distributions?
- Limited diversity of experimental setups, diversity of neural architectures - in the experiments, authors utilizes RBMs and ResCNNS, what will be the performance if we utilize more expressive architectures? As different architectures may exhibit different numerical sensitivities,  the generality of the findings remains uncertain.

---

> ### Author Rebuttal · Authors · 2026-03-30
>
> **Strengths**
>
> We sincerely thank the reviewer for the positive assessment and for recommending our paper for acceptance. We are glad that the practical insights of our work were appreciated. Indeed, a central goal of this paper is to equip researchers with concrete tools and guidelines for conducting VMC calculations in mixed precision in a controlled and reliable manner, and we are pleased that this contribution came through clearly.
>
> **Weakness 1**
>
> We thank the reviewer for this question, which gives us the opportunity to clarify the motivation behind the Gaussian assumption. Reduced-precision errors are deterministic at the level of individual operations and thus do not admit a probabilistic description in isolation. However, we model the aggregate error at the output of the neural network, which is the cumulative result of a large number of such operations. In this regime, the central limit theorem justifies treating the combined error as approximately Gaussian, provided the individual contributions are sufficiently independent. We further validate this choice empirically, finding good agreement between the Gaussian model and the observed output error distributions across several NQS architectures.
>
> Regarding extensions to non-Gaussian or adversarial error distributions: while such settings are beyond the scope of the present work, we agree that this is an interesting direction. Moreover, the framework—and in particular the bound provided in Theorem III.2—is completely general and could be extended by replacing the Gaussian assumption with a heavier-tailed distribution, should empirical evidence call for it, in order to explicitly compute the expectation value.
>
> **Weakness 2**
>
> We thank the reviewer for raising this point. We would like to clarify that the primary aim of this work is not to provide an exhaustive empirical survey of mixed-precision effects across all NQS architectures, but rather to develop general, architecture-agnostic tools for assessing the impact of reduced precision on variational optimisation. In particular, our bound serves as a simple proxy: given any chosen architecture, it captures the propagated error at the network output and relates it to the Monte Carlo sampling noise, yielding a principled criterion for whether mixed-precision arithmetic will introduce noticeable perturbations in the optimisation.
>
> The bound can therefore be applied to more expressive architectures, such as deep transformers or autoregressive networks, without modification, and we consider the empirical validation on RBMs and ResNets as a proof of concept rather than an exhaustive benchmark. In a recent VMC work, we have successfully used mixed-precision sampling with Pfaffian wavefunctions.
>
> We also note that the practical viability of reduced precision in expressive architectures is supported by recent work (Empowering deep neural quantum states through efficient optimization by Ao Chen & Markus Heyl), in which a transformer was employed for VMC and single precision was found to be sufficient for obtaining highly accurate results. This is consistent with the picture suggested by our framework: as network depth and width grow, the aggregate rounding error may remain well below the sampling noise floor, making mixed precision a safe and efficient choice.
>
> **Question 1**
>
> We thank the reviewer for this question, and would kindly ask for clarification: could the reviewer specify what is meant by “correction rate”? If they are referring to the contraction rate $r$ of the MCMC kernel, we would like to point out that this is not a hyperparameter of the optimisation, but rather an intrinsic property of the sampling process that depends on the target distribution and the efficiency of the chosen MCMC scheme. As such, it is not directly tunable during optimisation.
>
> That said, the contraction rate is intimately related to the autocorrelation time $\tau$ of the Markov chain (with slow contraction implying large $\tau$), and our bound makes this dependence explicit: samplers with long autocorrelation times will be more sensitive to reduced-precision perturbations, while chains that thermalise rapidly will be comparatively robust. In practice, this suggests that the choice of MCMC scheme, and in particular its mixing efficiency, is an important consideration when deploying mixed-precision VMC, even if the contraction rate itself is not freely adjustable.
>
> **Question 2**
>
> We thank the reviewer for this question. This is indeed an interesting direction for future work; however, given that the Gaussian assumption is both theoretically motivated by the central limit theorem and empirically justified by our findings across several NQS architectures, we chose to focus on this case as the natural starting point for the regime considered in this work.

---

> > ### Author Rebuttal · Reviewer_FtF9 · 2026-04-03
> >
> > My concerns have been adequately addressed.

---

### Official Review · Reviewer_w2C3 · 2026-03-13

**Soundness:** 3
**Presentation:** 3
**Significance:** 3
**Originality:** 4
**Overall Recommendation:** 5
**Confidence:** 3

**Summary:**

The authors study a mixed-precision approach for neural quantum states within VMC. The main idea is to use reduced precision primarily in the MH sampling step, while retaining higher precision for the rest of the optimization pipeline. The authors derive theoretical bounds on the bias introduced by finite-precision perturbations in MH sampling, then validate the theory empirically. Results suggest that sampling in lower precision can provide meaningful speedups without noticeably degrading the final training performance in the examined cases.

**Compliance With Llm Reviewing Policy:**

Affirmed.

**Final Justification:**

Thanks for the reply. I have no more questions, and I would like to keep the score as accept.

**Key Questions For Authors:**

1. Since the paper emphasizes speedups in sampling, I am curious to know the end-to-end wall-clock improvement for full VMC training, not just the isolated sampling kernel. How much total training acceleration is obtained in practice when all components are included?

**Limitations:**

Yes

**Strengths And Weaknesses:**

Strengths:
1. The study addresses a practically important systems-level bottleneck in NQS/VMC through a combination of theory and experiments, rather than relying solely on empirical optimization claims.
2. The implementation strategy is fairly clever and practical. The authors keep a double-precision copy for updates and use a lower-precision copy only for sampling.

Weaknesses:
1. The method accelerates the sampling portion, but the paper itself notes that larger end-to-end gains may require accelerating other expensive components, such as local-energy evaluation and preconditioning. This somewhat limits the overall practical impact if sampling is not the dominant bottleneck in a target workload.

---

> ### Author Rebuttal · Authors · 2026-03-30
>
> **Strengths**
>
> 1. The study addresses a practically important systems-level bottleneck in NQS/VMC through a combination of theory and experiments, rather than relying solely on empirical optimization claims.
> 2. The implementation strategy is fairly clever and practical. The authors keep a double-precision copy for updates and use a lower-precision copy only for sampling.
>
> We thank the reviewer for the positive assessment and for recommending our work for acceptance. We particularly appreciate the recognition of our effort to complement empirical results with a theoretical analysis, as well as the acknowledgment of the practicality of our mixed-precision implementation strategy.
>
> **Weaknesses**
>
> 1. The method accelerates the sampling portion, but the paper itself notes that larger end-to-end gains may require accelerating other expensive components, such as local-energy evaluation and preconditioning. This somewhat limits the overall practical impact if sampling is not the dominant bottleneck in a target workload.
>
> We agree with the reviewer that our current work focuses exclusively on accelerating the sampling component of VMC, and that this does not by itself guarantee maximal end-to-end speedups when other stages, such as local energy evaluation or preconditioning, dominate the computational cost.
>
> However, rather than viewing this as a limitation, we consider it a deliberate scoping choice and a starting point for a broader research direction. In particular, in Appendix I: *Effects on Other Parts of VMC*, we explicitly take initial steps toward extending mixed-precision techniques beyond sampling. There, we discuss the key numerical challenges associated with low-precision computation of local energies and gradients, and introduce diagnostic tools, such as analyses of gradient dynamic range and the condition number of the $S$ matrix, to characterize these issues.
>
> Our preliminary experiments indicate that naively applying low precision to these components leads to significantly more difficult optimization, often exhibiting bias and numerical instabilities. We attribute this behavior to the ill-conditioning of the $S$ matrix in certain regions of parameter space. These findings suggest that extending mixed-precision methods to the full VMC pipeline is nontrivial and requires a careful, dedicated treatment, which we believe is better suited for follow-up work. Including a complete treatment within the current paper would have compromised clarity, particularly given the space constraints of an 8-page ICML submission.
>
> At the same time, we emphasize that sampling is not a negligible component in practice: in our experiments, it accounts for approximately $20\text{-}30$ percent of the total runtime. This fraction will grow for larger system sizes, since the constant cost of inverting the S-matrix will contribute less to the overall runtime than the sampling and local energy calculation. Furthermore, for the other systems we are currently investigating, such as continuous-space VMC and lattice fermions with non-local proposals, sampling can become the dominant computational bottleneck. In such settings, our approach provides a natural and potentially impactful foundation for exploiting low-precision arithmetic.
>
> **Question**
>
> Since the paper emphasizes speedups in sampling, I am curious to know the end-to-end wall-clock improvement for full VMC training, not just the isolated sampling kernel. How much total training acceleration is obtained in practice when all components are included?
>
> We thank the reviewer for this question. As previously stated, the end-to-end speedup depends on the specific system under consideration, as the fraction of total runtime occupied by the sampling step varies across architectures and physical models. For the experiment in Figure 3 of the main text, we measured total training speedups of $2.057\times$ for `f32`, $2.296\times$ for `f16`, and $2.257\times$ for `bf16`. We will include these results in the revised manuscript.

---

> > ### Author Rebuttal · Reviewer_w2C3 · 2026-04-02
> >
> > Thanks for the reply. I have no more questions, and I would like to keep the score as accept.

---

### Decision · Program_Chairs · 2026-04-30

**Decision:**

Accept (regular)

**Comment:**

This manuscript considers the use of mixed precision when doing sampling within the context of VMC (e.g., for neural quantum states). The focus on sampling, while reasonable, is nevertheless a limitation in realizing significant "end-to-end" speedups. Nevertheless, the manuscript does a reasonable job with its formulation of the mixed-precision scheme (both theoretically and empirically) and illustrates its effectiveness. Collectively, the reviews agreed that there is a contribution here, albeit one of limited scope.